# TTDCapsNet: Tri Texton-Dense Capsule Network for complex and medical image recognition

**Vivian Akoto-Adjepong** [ID]*, **Obed Appiah, Patrick Kwabena Mensah, Peter Appiahene**

Department of Computer Science and Informatics, University of Energy and Natural Resources, Sunyani, Ghana

* vivian.akoto-adjepong@uenr.edu.gh

**Data Availability Statement:** The data underlying the results presented in the study are available from (https://github.com/vivianakotoadjepong/TTDCapsNet.git). Datasets supporting the findings of this study are publicly available at; fashion-

## Abstract

Convolutional Neural Networks (CNNs) are frequently used algorithms because of their propensity to learn relevant and hierarchical features through their feature extraction technique. However, the availability of enormous volumes of data in various variations is crucial for their performance. Capsule networks (CapsNets) perform well on a small amount of data but perform poorly on complex images. To address this, we proposed a new Capsule Network architecture called Tri Texton-Dense CapsNet (TTDCapsNet) for better complex and medical image classification. The TTDCapsNet is made up of three hierarchic blocks of Texton-Dense CapsNet (TDCapsNet) models. A single TDCapsNet is a CapsNet architecture composed of a texton detection layer to extract essential features, which are passed onto an eight-layered block of dense convolution that further extracts features, and then the output feature map is given as input to a Primary Capsule (PC), and then to a Class Capsule (CC) layer for classification. The resulting feature map from the first PC serves as input into the second-level TDCapsNet, and that from the second PC serves as input into the third-level TDCapsNet. The routing algorithm receives feature maps from each PC for the various CCs. Routing the concatenation of the three PCs creates an additional CC layer. All these four feature maps combined, help to achieve better classification. On fashion-MNIST, CIFAR-10, Breast Cancer, and Brain Tumor datasets, the proposed model is evaluated and achieved validation accuracies of 94.90%, 89.09%, 95.01%, and 97.71% respectively. Findings from this work indicate that TTDCapsNet outperforms the baseline and performs comparatively well with the state-of-the-art CapsNet models using different performance metrics. This work clarifies the viability of using Capsule Network on complex tasks in the real world. Thus, the proposed model can be used as an intelligent system, to help oncologists in diagnosing cancerous diseases and administering treatment required.

## Introduction

The leading cause of mortality worldwide in 2020 is cancer which accounted for around 10 million fatalities, or about one in every six [1]. In 2020, 2.3 million women worldwide were

MNIST: https://www.kaggle.com/datasets/zalando-research/fashionmnist CIFAR-10: https://www.cs.toronto.edu/~kriz/cifar.html Breast cancer: https://www.kaggle.com/datasets/ambarish/breakhis Brain tumor: https://www.kaggle.com/datasets/masoudnickparvar/brain-tumor-mri-dataset.

**Funding:** The author(s) received no specific funding for this work.

**Competing interests:** The authors have declared that no competing interests exist.

affected by breast cancer, which claimed 685 000 of them. Breast cancer, which had been discovered in 7.8 million women over the preceding five years, was the most prevalent cancer in the world as of the year 2020. More than any cancer type, more women had DALYs; lose disability-adjusted life years, due to breast cancer than any cancer type. Even though the incidence rates increase as people get older, breast cancer affects women after adolescence in every place on earth [2]. Comparing breast cancer to other cancer types, the mortality rate from breast cancer is particularly high [3]. Also, one of the most lethal and serious tumors in both adults and children is the brain tumor [4]. Tumors of the brain and spinal cord are collections of aberrant cells that have proliferated uncontrollably in the brain or spinal cord. By 2023, there will be 25,050 diagnoses of malignant tumors of the spinal cord and the brain (10,880 in women and 14,170 in men). If benign (non-cancerous) tumors were taken into account, these figures would be significantly higher. In 2023, brain and spinal cord cancers will claim the lives of about 18,280 persons (10,710 men and 7,570 women) [5].

A certified oncologist is responsible for the detection of such malignant diseases [6, 7]. Early detection and treatment of certain malignant disease types can be very successful. This is because it aids in creating a patient's treatment plan, which is crucial for oncologists and patients alike. One or more non-invasive and invasive procedures can be used to diagnose malignant disorders. Some of such procedures are; Breast exams, imaging tests (such as Magnetic Resonance Imaging (MRI)), obtaining a breast cell sample for analysis (biopsy), and another testing necessary for the diagnosis of breast cancer [8]. Additionally, a neurological examination, imaging tests (such as an MRI), a biopsy in which abnormal tissue is collected and tested, hearing, vision, and neurological tests, evoked potentials, electroencephalography (EEG), a neurocognitive assessment, etc. are all necessary for the diagnosis of brain tumors [9]. These processes necessitates a careful, in-depth analysis by specialist who may not be found everywhere, and when even found at a facility, the processes are time-consuming.

Deep learning models [10, 11] have been applied to many domains including medical health, to aid specialists, and make the classification of such diseases (malignant) much easier. They must increase performance on complicated images in terms of convergence, accuracy, flexibility, robustness, and complexity before it can be helpful to oncologists in carrying out the procedure of malignant illness identification effectively. CNNs are well-known and frequently used deep learning models because of their propensity to learn relevant and hierarchical features through their feature extraction technique with convolution structures [12]. They have been applied in the medical field, in diagnosing diseases such as Acne and Rosacea [13–15], brain tumors [16–18], breast cancer [19–21], etc. However, the availability of enormous volumes of data in various variations is crucial for their performance. This is problematic for the medical industry because, data generation and annotation are difficult tasks that can raise privacy concerns [22]. The pooling layer has been blamed for further CNN flaws [23]. The extracted features are translationally invariant locally when a convolution layer and a pooling layer are used, which causes them to loose a lot of information as they go deeper. As a result, multiple data augmentation techniques are needed to help CNN generalize well to other points of view. However, these augmentations are time-consuming and labor-intensive [24].

The idea of "capsules" was first proposed to address the issues of CNN [23, 25]. Capsule networks as opposed to CNNs need less training data and are resistant to uneven class distributions and changes in spatial orientation. As a result, capsule networks are a good choice for diagnosing medical images. Capsule networks can classify medical images without the need for data augmentation, but by learning to infer pose parameters from images [24, 26]. Even though CapsNet is resilient to perturbation and performs well on the majority of image classification tasks, they still have certain drawbacks [27]. As a result, they perform poorly on complex images [28] (i.e., varied background images), including CIFAR-100, CIFAR-10,

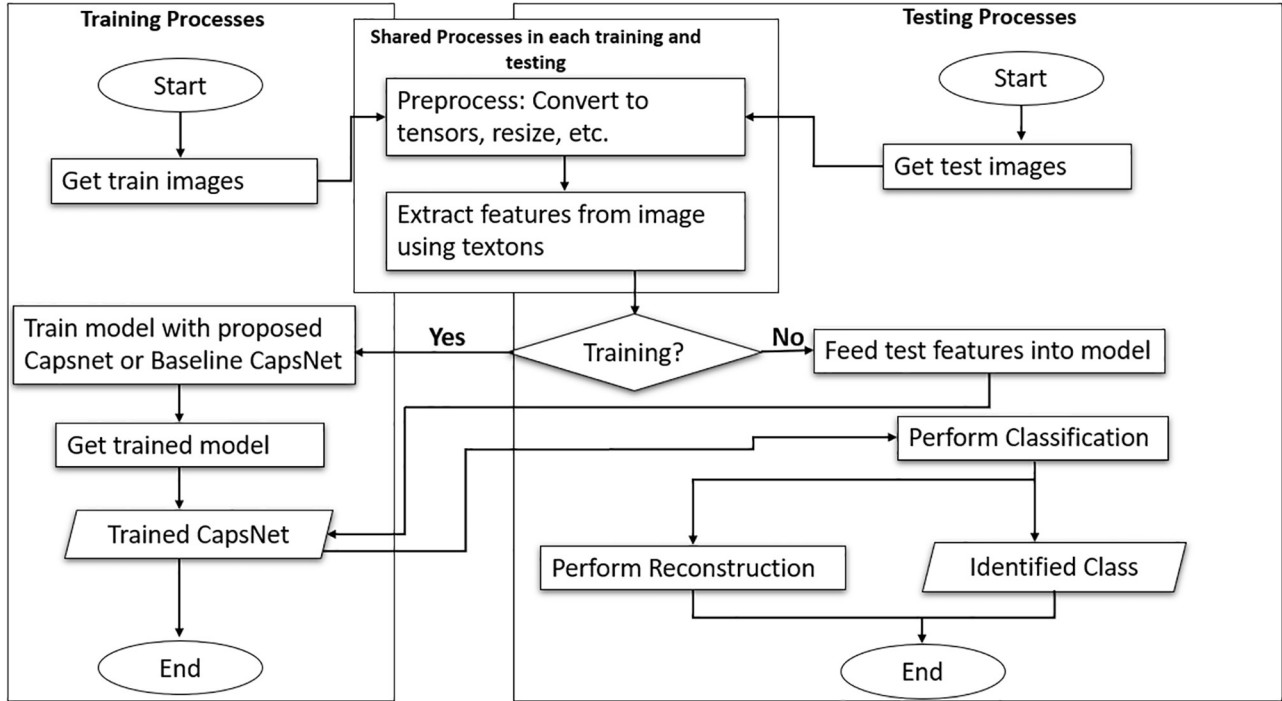

**Fig 1. Diagram of the work flow used in this study.**

multilabel and medical images because, they have the "crowding" problem. The capsule network strives to take into account every element of the image and is sensitive to the image background. The network performance may suffer as a result of this characteristic when considering detailed malignant images. The performance of CapsNet is significantly impacted by the encoder network's ineffective feature extraction [29]. Therefore, there is a need to improve the existing capsule network algorithm to help solve the problem of crowding, in order to classify complex images well or with precision.

This paper utilizes the dynamic routing algorithm of Capsule Network by incorporating a texton layer [30] to extract essential features, which are passed onto an eight (8)-layered block of dense convolution that further extracts features to improve the texture, color, and spatial recognition capabilities of Capsule Network, by enabling the decision of crucial features, and the needed coupling coefficients to be decreased in order to improve the hierarchical relationship among capsules that are related closely for better classification of complex images such as medical images. Fig 1 depicts the workflow adopted for the proposed work, which depicts the testing and training processes. The TTDCapsNet proposed, addresses the issue of crowding in capsules, and outperformed both the conventional CapsNet and some state-of-the-art models that can be found in literature on medical images and other complex images (e.g., CIFAR-10) according to experimental results on fashion-MNIST, CIFAR-10, Breast Cancer, and Brain Tumor datasets. Additionally, the suggested model fared better than the baseline model in terms of flexibility and convergence speed. For the detection of medical disorders like malignant diseases, a model's adaptability and capacity to generalize on unknown or distorted data are essential. Thus, the proposed model can act as an intelligent tool to help doctors identify malignant conditions and administer the required treatments. This paper's main contributions are as follows:

1) An improved, flexible and robust Capsule Network which incorporates texton and eight (8)-layered blocks of dense convolution for effective feature extraction is proposed.

2) Excellent color, edge, and texture extraction algorithms texton and dense convolution were suggested for better feature extraction, to help in identifying affected parts of medical images using Capsule Network.

3) A comparative analysis was conducted to evaluate the proposed model with other deep learning models, using metrics such as specificity, sensitivity, precision, accuracy, and others.

The organization of this article's structure is as follows; from the introduction, the next section focuses on related works on complex images and cancerous disease detection (breast cancer and brain tumor), the proposed methods follows the related works, then the results and discussion section follows the proposed methods section, and the last section focuses on the conclusion and future work.

## Related works

One of the domains of capsule network research that has garnered more attention is complex image detection. To enhance the performance of capsules on complex images, a number of proposals have been made, including those that involve ensemble averaging [31], stacking of more capsule layers, scaling factor changing for reconstruction, parallelizing capsule layers [32], increase in the number of layers in the primary capsule, and many others. In the detection of complex images, CNNs continue to outperform CapsNets.

This review starts with capsule networks proposed for the recognition of open datasets for deep learning models, CIFAR-10, Fashion MNIST, etc., and other datasets. Dynamic routing-based shallow CapsNet with a normalizer and a unique squashing function was proposed by Mensah and others [33]. Three (3) publicly accessible datasets, the fashion-MNIST, CIFAR-10, and tomato datasets were used to evaluate the model. For CIFAR-10 and fashion-MNIST, validation accuracy of 75.75%, and 92.70% respectively was achieved. By adding a feature deconstruction module to extract rich features and a multi-scale module for extracting essential features in capsules at the low level for medical image classification, Zhang and colleagues suggested an improved capsule network. The model was assessed using the datasets from PCam, MNIST, and CIFAR-10, and achieved validation accuracies of 88.23%, 99.65%, and 82.313%, respectively [34]. A study into CNN and CapsNet performance on images that are complex were conducted by Xi and colleagues. The development of 64 capsule layers was compared to convolutional layers that were ensembled and trained with CIFAR-10 and MNIST. On CIFAR-10, a test accuracy of 64.67% was attained [31]. Abra and colleague proposed a capsule network variant, using an amplifier algorithm and an exponential squash algorithm. The suggested approach was tested on four (4) datasets: CIFAR-10, eye disease, fashion-MNIST, and ODIR with validation accuracy rates of 84.56%, 89.02%, 93.76%, and 87.27% respectively [35]. A multilane capsule network with stringent squash is a new structure suggested for complex images. The model's test accuracies were 98.42%, 92.63%, 77%, and 76.79%, on MNIST, fashion-Mnist, affNIST, and CIFAR10 datasets respectively [32]. In 2018, Xiang and colleagues implemented a capsule network that is multi-scaled. The model proposed extracts feature that are multi-scaled and hierarchical features are also encoded to a primary capsule that is multi-dimensional. The network achieved test accuracies of 75.70% and 92.70% when applied to CIFAR-10 and fashion-MNIST datasets respectively [36]. Goswami in 2019, suggested a model in which the primary capsule layer uses residual blocks. When trained on CIFAR-100

and CIFAR-10, the model achieved an accuracy of 78.54% on CIFAR-10 [37]. Convolutional Fully Connected (CFC) CapsNet, which Shiri and Baniasadi proposed, is a more functional network with training and testing carried out more quickly and achieves a marginally better accuracy compared to the conventional CapsNet. Testing the model on MNIST, Fashion-MNIST, CIFAR-10, and SVHN datasets yielded validation accuracies of 99.64%, 92.86%, 73.15%, and 93.29%, respectively [38]. Zhao and others in 2019 suggested a technique for capsule networks' quick inference. After training, routing coefficients are accumulated by the suggested model to create a master routing coefficient, that is subsequently utilized to make inferences. The model significantly increased inference speed by replacing the routing-by-agreement loop iterations approach with a multiplication operation that uses a single matrix. The model's validation accuracies were 99.43%, 91.52%, and 70.33% respectively for MNIST, fashion-MNIST, and CIFAR 10 [39]. An inverted dot-product routing technique was proposed by Tsai and others. Concurrent iterative routing was introduced using this algorithm to replace sequential iterative routing. On CIFAR-10, the suggested model obtained 82.55% test accuracy [40]. DCNet and DCNet++ were suggested by Phaye and colleagues. By substituting densely connected convolutions for the typical convolutional layers in the two suggested models, the CapsNet was modified. On Fashion-Mnist, MNIST, SmallNORB, SVHN, Brain Tumor, and CIFAR-10 Datasets, the two models were assessed. The validation accuracies of the different datasets attained by the DCNet model are 94.64%, 99.75%, 94.43%, 95.58%, 93.04%, and 82.63%, and by the DCNet++ model at 94.65%, 99.71%, 95.34%, 96.90%, 95.03%, and 89.71%, respectively [41]. To transform to values that are normalized, where routing coefficients indicate probabilities of assignment among adjacent layer capsules, Zhao and colleagues suggested a model using max-min normalization in capsules. The model's accuracy was 75.92%, 93.09%, 92.07%, 95.42%, and 99.55% respectively on CIFAR-10, bMNIST, fashion-MNIST, rMNIST, and MNIST [42]. Ozan and friends also used quaternions to design quaternion CapsNet that represents the transformation and pose of capsules. 82.21% and 90.26% accuracies were achieved on CIFAR-10 and fashion-MNIST [43]. Ding and friends suggested residual CapsNet that uses maxpooling and routing algorithm that does group reconstruction. On SVHN and CIFAR-100 datasets, the model achieved accuracies of 98.64%, and 76.21% respectively [44]. Abra Ayidzoe and friends proposed a less sophisticated but resilient variation of a capsule network with good feature extraction abilities. To know the image's semantic and structural features, the model makes use of the benefits of blocks of custom-designed preprocessing and Gabor filters. According to experimental findings, the suggested model was able to correctly identify complex images from kvasir-dataset-v2, fashion-MNIST, CIFAR-100, and CIFAR-10 datasets with test accuracies of 91.50%, 94.78%, 68.17%, and 85.24% respectively [45]. Chang and team innovatively crafted a capsule-oriented network architecture, introducing modifications to the Squash function and refining the dropout implementation. Thorough experimentation was undertaken on three widely-used datasets—MNIST, affNIST, and CIFAR10—yielding accuracies of 99.73%, 81.71%, and 76.79%, respectively [46]. Also, Sun and colleagues introduced an innovative dense capsule network called DenseCaps, which utilizes dense capsule layers. The architecture comprises three Dense Capsule Blocks to facilitate the reuse of features and multi-scale feature capsule processing calculation of loss. The model demonstrated notable performance with an accuracy of 94.93% on fashion-MNIST and 89.41% on CIFAR-10 [47]. Through the incorporation of the Convolutional Capsule Layer (Conv-Caps Layer), Xiong and team enhanced CapsNet, leading to a significant boost in its overall performance. This enhancement resulted in validation accuracies of 81.29% for CIFAR-10 and 99.84% for MNIST datasets [48]. Do and colleagues presented a multi-lane capsule network (MLCN) designed for parallel processing, delivering notable accuracy with reduced costs. The MLCN comprises distinct parallel lanes, each contributing to a specific

result dimension. Evaluation of the model on fashion-MNIST and CIFAR-10 datasets demonstrated validation accuracies of 92.63% and 75.18%, respectively [49]. CHENG and colleagues introduced the Complex-valued Diverse CapsNet (Cv-CapsNet++), which involves a three-stage process. Initially, a restricted dense complex-valued subnetwork is utilized to acquire multi-scale complex-valued features. Subsequently, these features are encoded into complex-valued primary capsules in the second stage. Lastly, in the third stage, they extended the dynamic routing algorithm to the complex-valued domain, employing it to integrate real- and imaginary-valued information from the complex-valued primary capsules. The model achieved validation accuracies of 94.40% on Fashion-MNIST and 86.70% on CIFAR-10 datasets [50]. Shiri and her team substituted the second convolutional layer with a fully connected (FC) layer and directed the result of the initial convolutional layer into another FC layer. The FC layer's output was subsequently transformed to generate Primary Capsules, serving as inputs for the dynamic routing algorithm. Notably, the model attained validation accuracies of 88.84% and 67.18% on the fashion-MNIST and CIFAR-10 datasets, respectively [51]. Huang and Zhou introduced a capsule network with a dual attention mechanism, referred to as DA-CapsNet. In this model, the initial attention layer follows the convolution layer, while the subsequent attention layer is incorporated after the primary capsule layer. The model demonstrated notable validation accuracies of 93.98% and 85.47% on the fashion-MNIST and CIFAR-10 datasets respectively [52]. Luo and his team introduced R-CapsNet, drawing inspiration from the impact of a compact convolution kernel. This design serves as an expansion of the initial CapsNet, featuring four convolutional layers and one fully connected layer. During testing on the fashion-MNIST and CIFAR-10 datasets, the model demonstrated validation accuracies of 93.89% and 81.2%, respectively [53].

Modern capsule network models often obtain accuracies on CIFAR-10 that range from 65% to 82%. Our model so significantly outperforms the state-of-the-art, providing a contribution in this field. These studies [27, 54] can be consulted by interested readers who want to know more about the field, advantages, and disadvantages of capsule networks.

Other authors proposed CapsNet and other deep-learning models in order to identify malignant conditions for breast cancer and brain tumor detection. For breast cancer detection, independent component analysis (ICA) on a decision-making support system for breast cancer was investigated by Mert and colleagues. They demonstrated how the Radial Bases Function Neural Network (RBFNN) classifier becomes more distinctive with an increase in accuracy from 87.17% to 90.49% when given a one-dimensional feature vector from (ICA) [55]. Breast cancer categorization using the Inception Recurrent Residual Convolutional Neural Network (IRRCNN) model was proposed by Alom and friends. Residual Network (ResNet), Inception Network (Inception-v4) and Recurrent Convolutional Neural Network (RCNN) combined their strengths to form IRRCNN. Without data augmentation, the average model testing accuracy was 95.15%, whereas it was 97.22% when data augmentation was included [56]. A comprehensive framework containing color texture features, multiple classifiers, and a voting mechanism was proposed by Gupta and Bhavsar, who reported an average recognition rate for breast cancer categorization of about 87.53%. Ensemble classifiers, discriminant analysis, the nearest neighbor classifier, decision tree, and SVM were all utilized in this application [57]. Particle Swarm Optimized Wavelet Neural Network (PSOWNN) was utilized by Dheeba and friends to suggest a new method. Digital mammograms were used for the detection of breast anomalies in this model. The suggested abnormality detection system was built by the extraction of Laws textural energy values from mammograms and the classification of the concerned regions using a pattern classifier. For accuracy, specificity, and sensitivity, the model achieved 93.671%, 92.105%, and 94.167% respectively [58]. George and colleages also suggested a model based on SVM and neural networks (NN) for recognizing breast cancer and

achieved 94% recognition accuracy [59]. According to Bayramoglu and colleagues, different-sized convolution kernels were utilized in a CNN to categorize cancer of the breast independent of magnification. They used multi-task CNN (MTCNN) and CNN models to perform patient-level categorization of cancer of the breast, with a reported recognition rate of 83.25% [60]. A deep learning-based technique called DeCAF was proposed by Spanhol and friends, it employs CNN that is pre-trained to obtain important features and used these features as input to a classifier. At the image level and the patient level, the technique obtained classification accuracy of 84.2% and 86.3% respectively [61]. A convolutional neural network that is deep (CSDCNN) and class structure-based was proposed by Han and others. Both at the imaging and patient level, this model performs at the cutting edge. For patient-level breast cancer categorization, an overall accuracy of 93.2% was recorded [62]. Deep feature extraction and Transfer learning techniques were utilized by Deniz and colleagues to modify a CNN model that is pre-trained. To extract important features, AlexNet and Vgg16, models were employed, and AlexNet is used for additional fine-tuning. Support vector machines were then used to classify the collected features. The model had a 91.37% accuracy rate [63]. On the BreakHis publicly accessible dataset for cancer of the breast categorization on histological images, Agarwal and others conducted performance research on a deep CNN and four well-known architectures based on CNN: ResNet 50, MobileNet, VGG19, and VGG16. VGG16 demonstrated the best performance with 80.52% recall, 85.21% f1-score, 92.60% precision, and 94.67% accuracy value [64]. For the purpose of classifying malignant and benign from breast cancer histopathology images, Zhang and others used a kernel that has a one-class principal component analysis (PCA) technique based on features that are hand-crafted. The accuracy obtained was 92% [65]. A stochastic down sampling unit, an SDC model and a stochastic upsampling unit for effectively retrieving detailed features, and a feature selection component for creating a useful feature map are all combined in Kashyap proposal of the SDRG model. The model had a 95.15% accuracy rate when tested against a dataset of breast cancer [66]. Anupama and others, employed CapsNet with Histology Images that are preprocessed and obtained an accuracy of 92.14% [67]. Several SMV-based methods were used by Kahya and others to identify breast cancer, with an Adaptive Sparse SVM (ASSVM) achieving an accuracy of 94.97% [68]. Iesmantas and Alzbutas also used CapsNet to classify breast cancer and got an accuracy of about 87% [69]. Results for both patch-based and image-based analysis are provided by a model suggested by Araújo and others. When evaluated on the BC Classification Challenge 2015 dataset, the technique based on CNN achieved roughly 77.8% classification accuracy for the 4-class analysis and 83.3% classification accuracy for the 2-class analysis [70]. MyResNet-34, a CNN based on residual learning, was proposed by Hu and others for the categorization of malignant and benign tissues. The bias brought on by manually choosing the reference image was removed by an algorithm that automatically generated the target image for stain normalization. For image-level classification on BreakHis dataset, the model attained a classification accuracy of about 91% [71]. Also, a more advanced capsule network was suggested by Khikani and others, this extracts multi-scale characteristics utilizing the Res2Net block and four extra convolutional layers. Furthermore, the Res2Net block and small convolutional kernels used in the proposed technique result in fewer parameters. The suggested model had an accuracy of 95.6% and a recall of 97.2% after being trained and tested on the publically available BreakHis dataset [72]. Gheshlaghi and colleagues used Generative Adversarial Network (ACGAN) to produce images that are realistic along with their labels and performed categorization of breast cancer histopathological images using deep CNN classifiers that are transfer learning based. The effectiveness of the ACGAN was evaluated and the model obtained a categorization accuracy of 90.15% [73]. A study based on an AlexNet-like model using several fusion approaches (such as max, product, and sum) for the patient and image-level categorization of cancer of the breast

was reported by Spanhol and colleagues. This study used the max fusion method to classify patients at the patient level and recognize images with a recognition accuracy of 85.6%, and 90% respectively [74]. A deep feature fusion and improved routing (FE-BkCapsNet) architecture was suggested by Wang and others, this makes use of the strengths of CapsNet and CNN. A published dataset called BreaKHis was used to evaluate the suggested approach. Experimental results for the dataset's images' accuracy at various magnifications were (400X: 93.54%, 200X: 94.03%, 100X: 94.52%, and 40X: 92.71%) [75].

For brain tumors, in order to improve MRI quality and provide a discriminative feature set, Ayadi and others suggested a method that makes use of dense sped-up robust features, normalization, and histogram gradient techniques. During the classification phase, a support vector machine was used. The proposed approach was evaluated using a dataset of brain tumors. This approach resulted in an accuracy of 90.27% [76]. With the use of the entire volumetric T1-Gado MRI dataset, Mzoughi and others developed an efficient and automated deep multi-scale CNN that has 3 dimensions (3D CNN) for classifying glioma brain tumors into High-grade gliomas (HGG) and Low-grade gliomas (HGG). The suggested model had an overall accuracy of 96.49% [77]. A hybrid approach employing CNN and Neutrosophy (NS-CNN), was proposed by Özyurt and others. The approach was used to categorize benign or malignant segmented tumor regions from brain tumor images. The accuracy of the suggested model was 95.62% [78]. In order to increase the focus of CapsNet, Afshar and colleagues suggested an improved CapsNet architecture for classifying brain tumors that incorporates the tumor coarse boundaries as additional inputs. The validation accuracy for the model was 90.89% [79]. Using a deep learning approach to segment tumor regions, with comprehensive augmentation of data, and a pre-trained CNN fine-tuned model for brain tumor grade classification, Sajjad and colleagues presented a modified (CNN) based multi-grade system for classifying brain tumor grades. The model's accuracy was 90.67% [80]. According to Adu and colleagues, an improved CapsNets with several convolutional layers and dilation to preserve image resolution and boost classification accuracy was proposed. The proposed system can guarantee an increase in CapsNets focus by inputting segmented tumor regions within the structure. This model's performance obtained an accuracy of 95.54% [81]. The CapsNet model presented by Afshar and colleagues is identical to that in the original publication in terms of nature, with the exception that it uses 64 feature mappings in the convolutional layer rather than 256 as in the original model. When tested on a dataset of breast cancer cases, the suggested model had an accuracy rate of 86.56% [82]. Afshar and others also introduced a boosted capsule network (also known as BoostCaps), that makes use of boosting approaches' capacity to accommodate poor learners, by steadily boosting the models. The BootsCaps architecture, according to the results, classified brain tumors with an accuracy of 92.45% [83]. A CNN model was proposed by Badža and Barjaktarovic for classing three distinct forms of brain tumors. The suggested model, which has a straightforward architecture akin to traditional CNN accurately classified 96.56% of the brain tumor MRI images in the dataset [84]. A capsule network (CapsNet) based framework for classifying different types of brain tumors was proposed by Adu and colleagues. Atrous expands receptive fields and preserves spatial information in the developed framework, whilst CLAHE applies a better histogram equalization that is adaptive (AHE) to improve images that were input. Utilizing segmented and whole-brain tumor datasets, the proposed technique was assessed. The experimental findings of the model provided accuracies (93.40% and 96.60%) and precisions (94.21% and 96.55%) for the images that were original and obtained accuracy (98.48% and 98.82%) and precisions (98.88 percent and 98.58 percent) for the augmentation of the two datasets [85]. In order to solve the classification challenge for brain tumour, Ayadi and colleagues suggested a new model that makes use of the CNN sequential model. The model has several layers and was designed to categorize MRI brain cancers. The model had a

94.74% accuracy rate [86]. A more universal dual-channeled deep neural network (DNN) architecture for classifying brain tumors was proposed by Bodapati and others. In the beginning, Xception and InceptionResNetV2 networks' convolution blocks are used to extract specific feature representations, which are then vectorized by the use of suggested pooling-based approaches. A proposed attention mechanism allows for greater emphasis on tumor regions and decreased emphasis on non-tumor parts, thereby assisting in identifying the tumor type found in the images. With and without data augmentation, the model attained an accuracy of 95.23% and 98.04%, respectively [87]. DCNet and DCNet++ were suggested by Phaye and colleagues. By substituting densely connected convolutions for the typical convolutional layers in the two suggested models, the CapsNet was modified. On Fashion-Mnist, MNIST, Small-NORB, SVHN, Brain Tumor, and CIFAR-10 datasets, the two models were assessed. For the brain tumor dataset, the DCNet model and DCNet++ model both achieved a validation accuracy rate of 93.04% and 95.03%, respectively [41]. By implementing an expectation-maximization (EM)-based dynamic routing algorithm and three fully connected layers, Goceri suggested a capsule network that can extract necessary characteristics from images and classify automatically brain tumors with 92.65% accuracy [88]. Afshar and others presented the Bayes-Cap, a Bayesian CapsNet architecture that can offer both the mean forecasts and entropy as a gauge of uncertainty in forecasting. According to the findings, accuracy can be increased by filtering out uncertain forecasts. The model's maximum accuracy was 73.9% with a CI of: (73.5%, and 74.4%) [89].

All the proposed models performed well on the various datasets. But for medical image diagnosis, there is a need for a more robust and efficient model for better diagnosis, hence this study aims to propose an improved, flexible and robust Capsule Network which incorporates texton and eight (8)-layered block of dense convolution for effective feature extraction, for better classification of malignant diseases.

## Proposed methods

Fig 1 is the adopted workflow used for this work.

### Capsule network

A capsule [28] is a collection of neurons, where the activity vector length denotes the probability of an entity existing, and the direction of the vector denotes the instantiation parameters. A CNN layer comes first in a capsule network, followed by a Primary Capsule (PC) layer, then Classification is handled by the Class Capsule (C) layer and Reconstruction is handled by the decoder network which is fully connected. The CNN layer does the feature extraction and the output is given as input into the PC layer. Each capsule $i$ (where: $1 \leq i \leq N$) present in the $l$ layer has activity vector $u_i \epsilon \ \mathbb{R}$, which in instantiation parameter form, encodes the spatial information. The $i^{th}$ lower-level capsule's output vector $u_i$, is given as input to all capsules in the $l + 1$ layer. $u_i$ will be received by layer $l + 1$ s' $j^{th}$ capsule, and the product $W_{ij}$ will be found. The product of activity vector $u_i$ and its weight matrix $W_{ij}$ is the primary capsule output $\hat{u}_{j|i} = W_{ij}u_i$. With the coupling coefficient $C_{ij} = \frac{exp(b_{ij})}{\sum_k exp(b_{ik})}$, which is the SoftMax of the logits $b_{ij} = b_{ij} + v_j \ .\hat{u}_{j|i}$, the primary capsule predictions are connected to a suitable class capsule. Upon agreement $a_{ij} = v_j \ .\hat{u}_{j|i}$ among a lower-level and a higher-level prediction, $C_{ij}$ get updated in the course of the dynamic routing process indicated in Algorithm 1. A higher-level capsule $j$ entire input is the weighed-sum of every vector predicted $\hat{u}_{j|i}$ of a specific primary capsule $i$ for a specific class capsule $j$, this is given by $s_j = \sum_{i=1}^{N} c_{ij} \ .\hat{u}_{j|i}$, which represents the

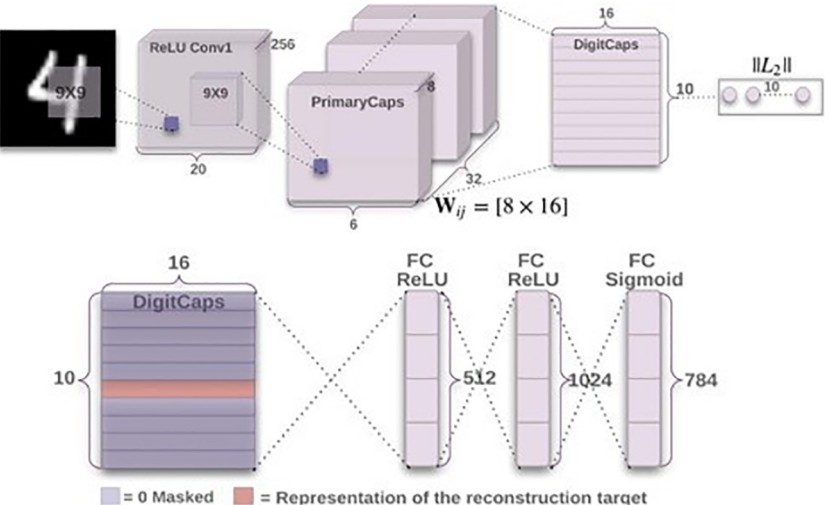

**Fig 2. Architecture of the baseline capsule network model [28].**

vector input to the $j^{th}$ capsule. A non-linear squashing function $v_j = \frac{||s_j||^2}{1+||s_j||^2} \frac{s_j}{||s_j||}$ is used to limit the output value of the Class Capsule between [0, 1], in order to prevent vector outputs from exceeding 1. Hence $v_j$ represents the output vector showing the probability of the vector between 0 and 1. In the last layer, a capsule labeled $k$ is connected with a loss $l_k$. The final layer contains Capsule $k$, which is associated with a loss denoted as $l_k$. This leads to a significant loss when capsules exhibit extended output instantiation parameters in cases where the entity being represented does not actually exist. The expression for the loss function $l_k$ is found in Eq 1 below;

$$L_k = T_k \ max(0, m^+ - ||v_k||)^2 + \lambda(1 - T_k) \ max(0, \ ||v_k|| - m^-)^2 \tag{1}$$

Where $T_k$ = 1 when class $k$ is active and 0 otherwise. Hyper-parameters λ, m-, m+ are set during the process of learning. On a variety of problems, CapsNets shown in Fig 2, have shown good performance [27].

**Algorithm 1**: **Dynamic Routing-by-Agreement**

**Procedure** ROUTING $(\hat{u}_{j|i}, r, l)$
**For** all capsule $i$ in layer $l$ and capsule $j$ in layer $(l + 1)$: $b_{ij} \leftarrow 0$
**For** $r$ iterations **do**
 **For** all capsule $i$ in layer $l$: $c_i \leftarrow SoftMax(b_i)$ ▷ softmax computes $c_{ij}$
 **For** all capsule $j$ in layer $(l + 1)$: $s_j \leftarrow \sum_{i=1}^{N} c_{ij} . \hat{u}_{j|i}$
 **For** all capsule $j$ in layer $(l + 1)$: $v_j \leftarrow squash(s_j)$ ▷ squash computes $v_j$
**For** all capsule $i$ in layer $l$ and capsule $j$ in layer $(l + 1)$:
$b_{ij} = b_{ij} + v_j . \hat{u}_{j|i}$
**return** $v_j$

## Texton detection

The texton hypothesis put forward by Julesz [90] forms the basis of the multi-texton histogram (MTH) [91]. Four different texton kinds are used by MTH to identify an image's microstructure. MTH uses a color histogram in RGB color space to retrieve image attributes, and the

Sobel operator to identify an image's edge orientation. It can serve as a description of color and texture and increases performance. Four steps make up the MTH stages. The first is to use the Sobel operator to detect the edge orientation. Secondly, the RGB color space of the image should then be quantized. Thirdly, use four separate textons to detect texton on the outcomes of quantized color and quantized edge orientation. The detection method is carried out via a two-pixel shift from left-to-right and from top-to-bottom. Color histograms and edge orientation histograms are produced as a result of the texton detection. Finally, a single histogram is created by combining the edge orientation and color histograms. The combined histogram is made up of features that include edge orientation and color features. Textons are typically defined as a collection of blobs that are distributed throughout the image and have a common attribute. However, the texton definition is still an issue. Jublesz offered a more thorough formulation of the texton theory [91], emphasizing the crucial distances among texture elements that are necessary for the computation of texton gradients. Only when the surrounding elements are in the neighborhood do textures form. However, the size of the elements affects this neighborhood. Pre-attentive discrimination is slightly hampered by considerably increased texture features in one orientation. The texton gradients found at the texture boundary lines rise if the elongated components are not jiggled in orientation. Because texton gradients only exist at texture borders, discrimination of texture can be improved with a small element size like 2x2 [92]. In MTH, four unique texton types are used on a 2x2 grid. Assign the four pixels the letters V1, V2, V3, and V4. The grid will generate a texton if the two pixels shaded in blue (as shown in Fig 3) have the same value. They are referred to as T1, T2, T3, and T4 accordingly. To find textons with a step length of 2 pixels, first, move the 2x2 block from top to bottom, and then, left to right across the color-indexed image C(x,y). The initial pixel values in the 2x2 grids are maintained if a texton is found. Otherwise, it will be worth nothing. The result then becomes a texton image, indicated by the letter T (x, y). Richer information is present in the four texton kinds utilized in MTH than that of TCM because the likelihood of two identically valued pixels co-occurring is higher than that of three or four identically valued pixels in a 2x2 grid [30]. Multi Texton Detection (MTD) approach is used to extract edge and color information for this model. Six different texton kinds are used to complete the texton detecting process [92]. The MTH method is the foundation for the texton detection in this study. The variation is based on adding on of two (2) distinct textons (f, g), hence making 6 textons instead of the original 4 (T1, T2, T3, T4, T5 and T6), as shown in Fig 3. The new textons are vertical right and horizontal bottom [92] (f and g textons). When pixels appear at the vertical right and horizontal bottom simultaneously, these textons aid in preventing information loss. The texton detection process using 6 textons is illustrated in Fig 4 and Algorithm 2.

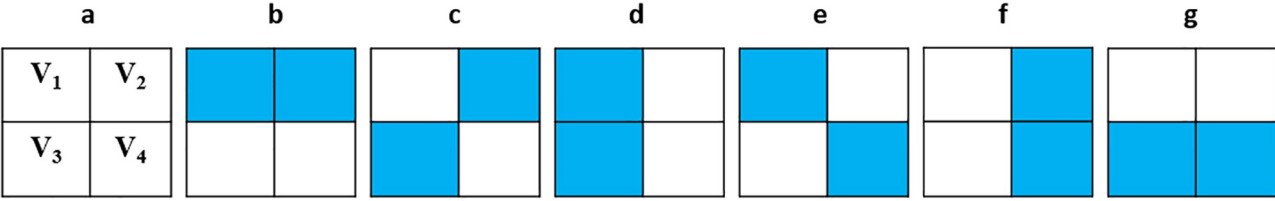

**Fig 3.** Six texton types used in texton detection process: (a) 2x2 grid; (b) T1; (c) T2; (d) T3; (e) T4; (f) T5; (g) T6.

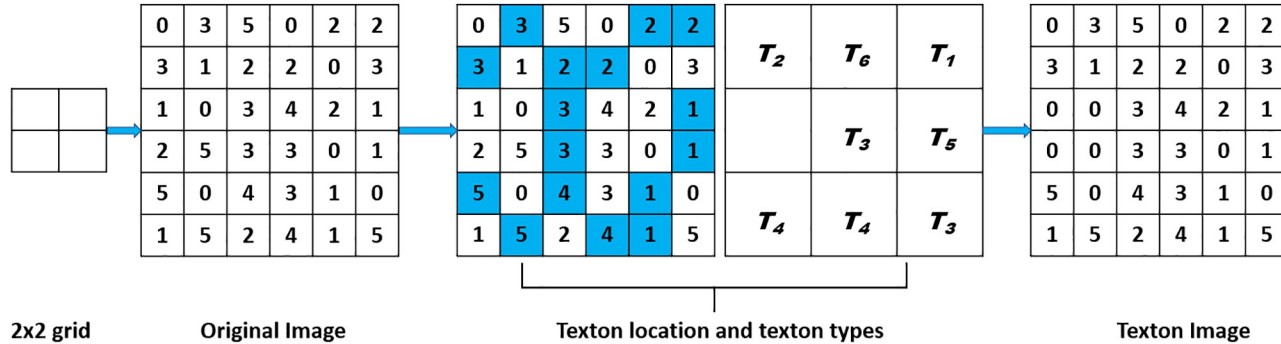

**Fig 4. Illustration of the texton detection process.**

**Algorithm 2**: **Texton detection algorithm using 6 textons** 6

```
img → image
numrows → number of rows in img
numcolumns → number of columns in img
for i = 0 to numrows: step 1
  for j = 0 to numcolumns: step 1
    if (img[i][j] equals img[i][j+1]) then
      keep original pixel values in img corresponding to T1 grid
    else if (img[i][j+1] equals img[i+1][j]) then
      keep original pixel values in img corresponding to T2 grid
    else if (img[i][j] equals img[i+1][j]) then
      keep original pixel values in img corresponding to T3 grid
    else if (img[i][j] equals img[i+1][j+1]) then
      keep original pixel values in img corresponding to T4 grid
    else if (img[i][j+1] equals img[i+1][j+1]) then
      keep original pixel values in img corresponding to T5 grid
    else if (img[i+1][j] equals img[i+1][j+1]) then
      keep original pixel values in img corresponding to T6 grid
  end for
end for
for i = 0 to numrows: step 1
  for j = 0 to numcolumns: step 1
    if (img pixel position is not detected by: T1, T2, T3, T4, T5 or
T6) then
      replace those image pixel values with zeros
end for
end for
return img
```

## Proposed model

The proposed Tri Texton-Dense CapsNet (TTDCapsNet) model (shown in Fig 5) consists of three texton detection layers, three eight-layered dense convolutional block, three primary capsule layers, four class capsule layers, and a decoder network.

The Tri Texton-Dense CapsNet (TTDCapsNet) is made up of three hierarchic blocks of Texton-Dense CapsNet (TDCapsNet) models, where a TDCapsNet model is designed and the intermediate representation is used as input into the next TDCapsNet which then generates another representation that is fed into the next TDCapsNet layer. A single TDCapsNet is a CapsNet architecture composed of a texton detection layer to extract essential features, an

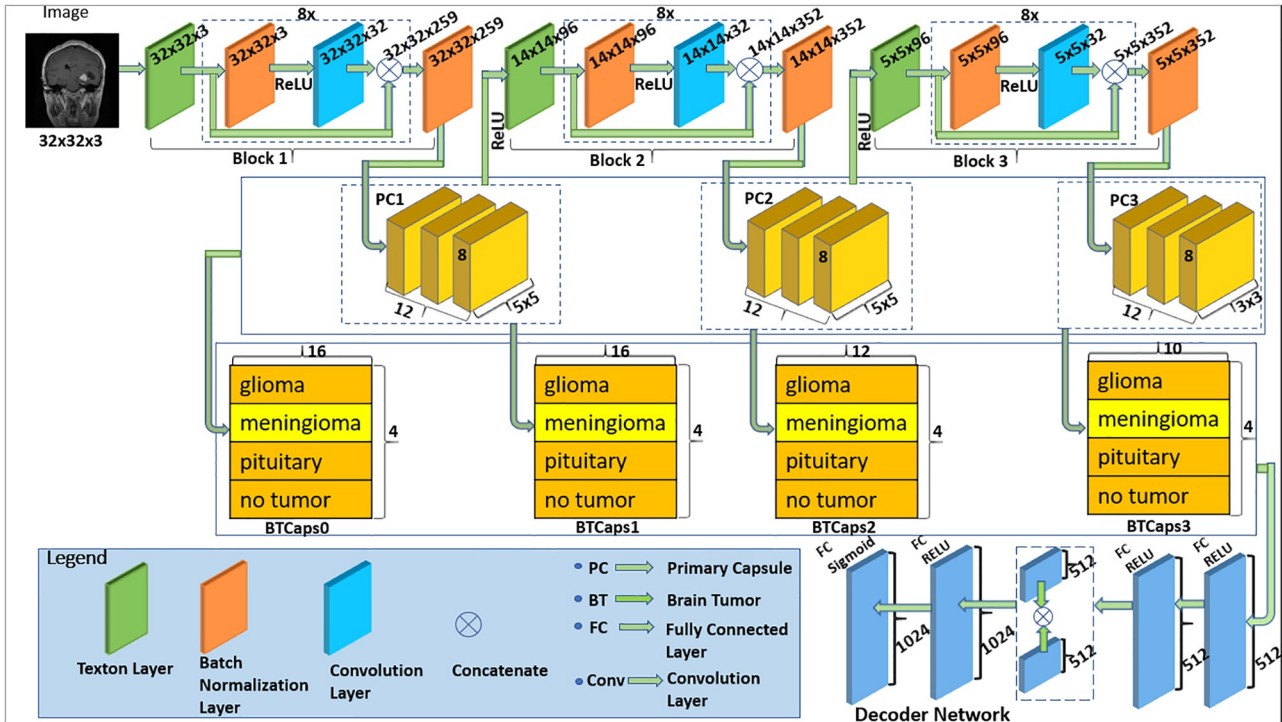

**Fig 5. Architecture of the proposed CapsNet model.**

eight (8)-layered block of dense convolution based on skip connections, a Primary Capsule and Class Capsule layer. The input sample is subjected to a texton layer for important feature extraction, then to 8 levels of convolution after applying batch normalization and ReLU activation. These new feature maps are then concatenated with feature maps from all previous layers which have different levels of complexity, and this serves as input into the Primary Capsules, further improving classification. Block 1 produces 32 x 32 x 259 feature maps which become input to the first Primary Capsule (PC1), and the resulting 14 x 14 x 96 feature maps also become input to the second TDCapsNet. Block 2 produces 14 x 14 x 352 feature maps which become input to the second Primary Capsules (PC2) and the resulting 5 x 5 x 96 feature maps also become input to the third TDCapsNet. Lastly, Block 3 produces 5 x 5 x 352 feature maps which become input to the third Primary Capsules (PC3). Each of the three primary capsules (PC1, PC2, PC3) is made up of 12 capsules with 8 dimensions or sizes (8D) and a stride of 2. The kernel size used for PC1 and PC2 is 5 x 5 and that of PC3 is 3 x 3. Following a squash activation layer, the routing algorithm receives these feature maps from each primary capsule. The TTDCapsNet model consists of 4 Class Capsules. 16D, 16D, 12D, and 10D, generated for BTCaps0, BTCaps1, BTCaps2, and BTCaps3 in the Class layer of which each further generates a 4D output vector, representing the 4 classes in the Brain Tumor dataset. The Class Capsule consisting of 16D for BTCaps0 is added by routing the concatenation of three (3) Primary Capsules (PC1, PC2, PC3) to create an additional Class Capsule layer. In order to learn integrated features from several layers of capsules by the model, that additional Class Capsule layer was included. The next 16D, 12D, and 10D Capsules are for the first, second, and third TDCapsNet of the model. The decoder is four fully connected layers having 512, 512, (for good reconstruction, there was a concatenation of the first two layers), 1024, and 3072 neurons respectively, which takes the Class Capsule layers output as its input. The model was trained

jointly, however, the losses of the four layers were back-propagated independently to prevent any imbalanced learning. The reconstructions were made using only 1 channel of the 4 Class Capsule layers of the image, which were concatenated during testing to build fifty-four (54D) final capsules for every one of the classification classes. In the suggested model, we down-sampled the entire image to a size of 32 x 32 before passing it. This model leverages the model by Phaye and colleagues [41].

## Datasets

On four datasets, namely fashion-MNIST [93], CIFAR-10 [94], breast cancer [95] and brain tumor [96], we conducted comprehensive experiments. Fig 6 shows sample images from breast cancer and brain tumor datasets.

fashion-MNIST: consists of 10,000 test images and 60,000 training images with a dimension of $28 \times 28 \times 1$. Ten (10) classes make up Fashion-MNIST: Top/ T-shirt: 0, trouser: 1, pullover: 2, dress: 3, coat: 4, sandals: 5, shirt: 6, sneakers: 7, bag: 8, and ankle boot: 9.

CIFAR-10: contains 60,000 32 x 32 colored images of which 10,000 are test images and, 50,000 are training images. CIFAR 10 has ten (10) classes: airplane: 0, automobile:1, bird:2, cat:3, deer:4, dog:5, frog:6, horse:7, ship:8, and truck:9.

Breast Cancer: Consists of 7908 histopathological images of the human breast, divided into two categories: benign tumors and malignant tumors. This makes benign: 0 and malignant: 1. The dataset was scaled to 32 x 32 x 3 and redistributed following an 80:20 leave-out strategy.

Brain Tumor: consists of 7022 images of MRI scans of the human brain that have been categorized into four (4) classes: glioma: 0, meningioma: 1, pituitary: 2, and no tumor: 3. The dataset was scaled to $32 \times 32 \times 3$ and redistributed using an 80:20 leave-out strategy.

## Experimental setup and performance evaluation

On a 64-bit Windows computer equipped with an NVIDIA GeForce RTX 2080 SUPER Graphic Processing Unit (GPU) with 8 GB GPU memory, 32 GB of system memory, and the CUDA 10.1 toolset, the suggested model was trained and tested. All coding was done in Keras with a TensorFlow backend. Adam optimizer with a learning rate set at 0.001 was used. All training were done using batches between 50 and 100, and the best model was preserved. In order to determine the loss, we used the margin loss $L_k$ found in Eq 1, from [28].

## Performance evaluation measures

The following are the classification performance metrics utilized in this study:

Validation Accuracy: the proportion of categories that were successfully categorized to all of the categories. We give an example of the overall validation accuracy attained across all experiments.

Loss: This metric displays the discrepancy between the model's prediction and the actual labels. Margin loss is employed in this test.

Confusion Matrix (CM): A detailed study of the number of images that were incorrectly or correctly classed is provided by the metrics obtained from the CM. The evaluation of precision, accuracy, specificity, sensitivity (recall), and many other metrics uses parameters;

True positive (TP): Correct prediction with positive actual value.

False positive (FP): Wrong prediction with positive actual value.

True negative (TN): Correct prediction with negative actual value.

False negative (FN): Wrong prediction with negative actual value.

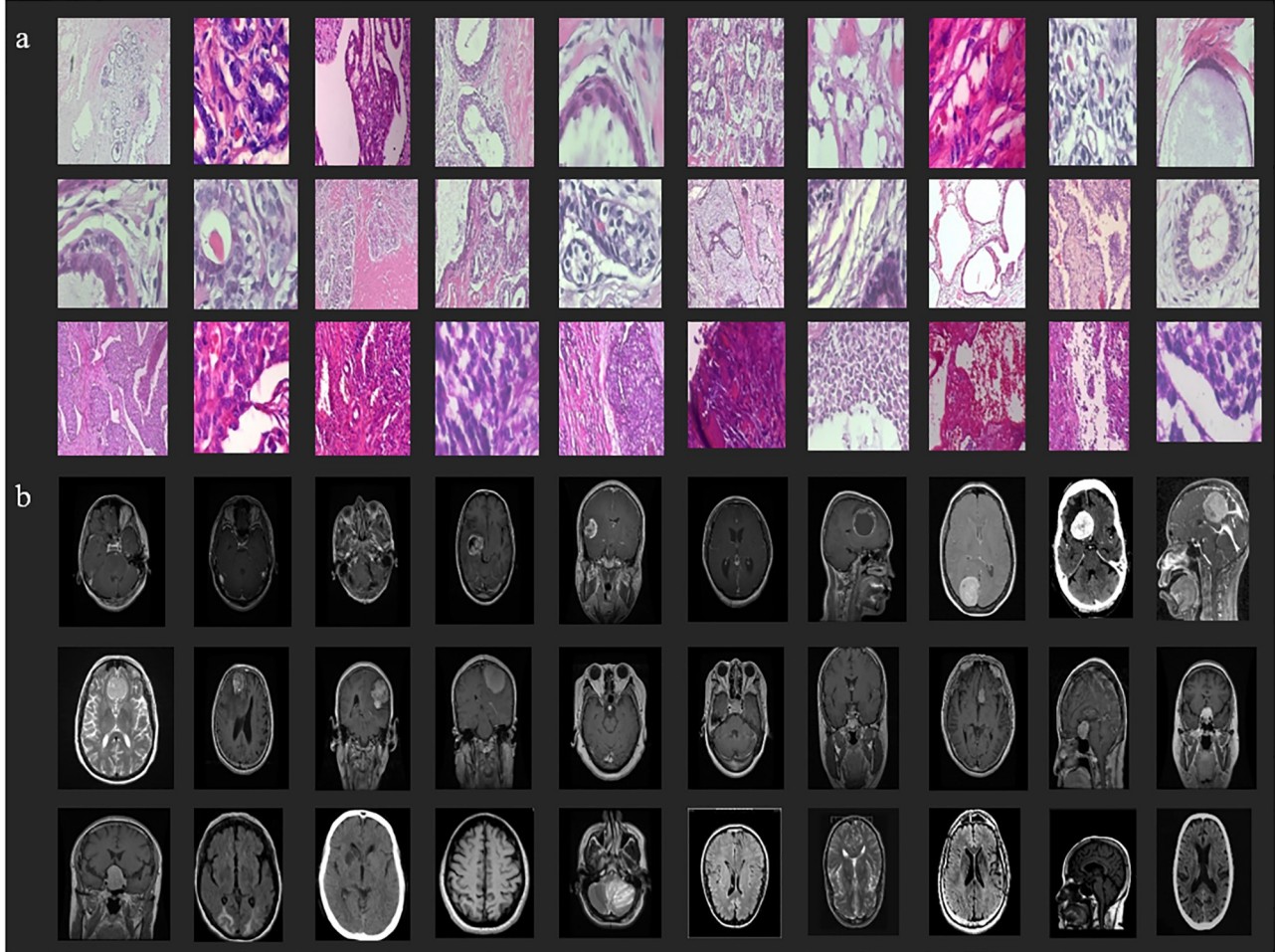

**Fig 6.** Sample images from the two medical image datasets; **a)** breast cancer **b)** brain tumor.

Accuracy (for each class): The ratio of items that were correctly classified (TP + TN) to all the items included in the test set (TP + FP + TN + FN)

$$Accuracy = \frac{TP + TN}{TP + FP + TN + FN} \qquad (2)$$

Precision: the number of images that was predicted as particular class and actually belongs to that class. (True positives to all predicted positives ratio)

$$Precision = \frac{TP}{TP + FP} \qquad (3)$$

Recall or Sensitivity: The number of images that is correctly predicted, of all the images in a particular class. (The proportion of true positives to all actual positive data points)

$$Sensitivity = \frac{TP}{TP + FN} \qquad (4)$$

Specificity: The proportion of true negatives to all of the data's negatives

$$Specificity = \frac{TN}{TN + FP} \tag{5}$$

Area under the curve (AUC): To assess how well the model performs on imbalanced datasets, the area under the receiver operating characteristic (ROC) and precision-recall curves (PR curve) are produced. The higher the values, the better the robustness of the model.

## Results and discussion

The performance of the proposed model on the four datasets utilizing validation accuracy, classification loss, and testing on unseen images (prediction) is thoroughly examined in this section. The outcomes are then contrasted with those from other models in the literature. The four datasets training and test graphs are shown. To account for unbalanced datasets, the performance of the model is further assessed using the Receiver Operating Characteristic (ROC) and Precision-Recall (P-R) curves. An ablation study is used to test the proposed model's robustness and to identify which component of the model improves performance the most.

### Performance evaluation

From Figs 7–10, the proposed model performed very well in terms of accuracy, since it obtained high validation accuracies of 94.90%, 89.09%, 95.01%, and 97.71% for fashion-MNIST, CIFAR-10, Breast Cancer, and Brain Tumor datasets, representing the correct predictions from the entire predictions made. This shows that, the proposed model generalizes very well on unseen data. Furthermore, confusion matrices of the proposed and baseline models are depicted in Figs 11 and 12. This indicates how many images were correctly and incorrectly identified. The diagonal cells highlighted in blue represent accurate predictions made by the model, including both true positives and true negatives. Misclassifications produced by the model are shown in white, both above and below the correct predictions. We calculated the

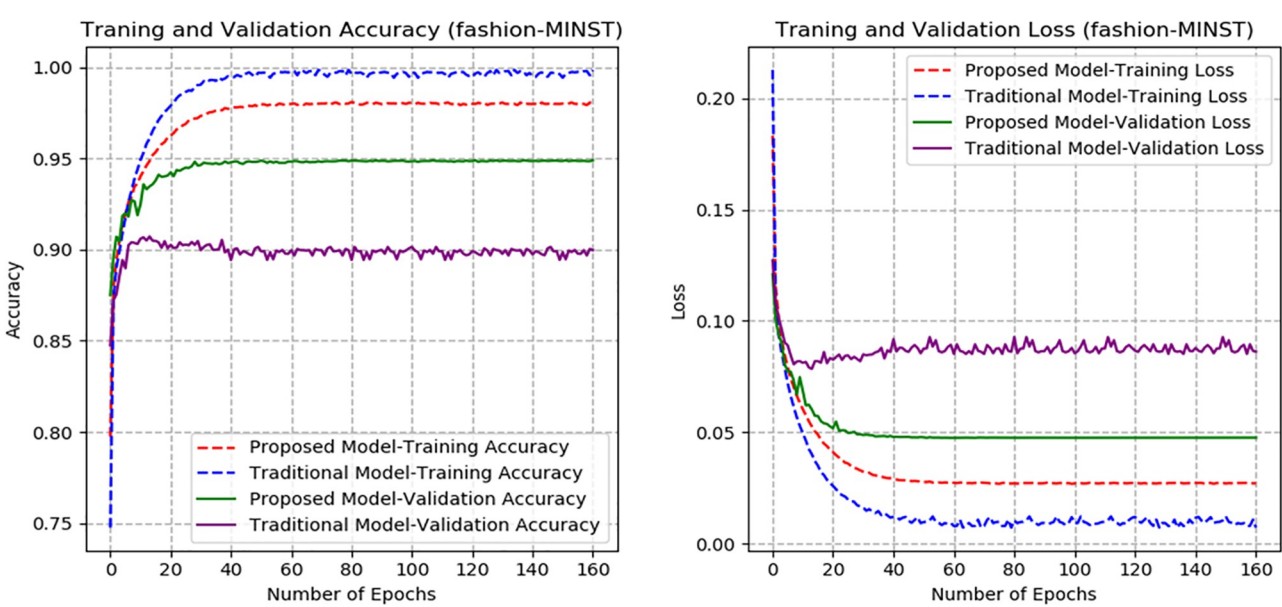

**Fig 7. Accuracy and Loss graphs of the proposed and baseline CapsNet models (fashion-MNIST).**

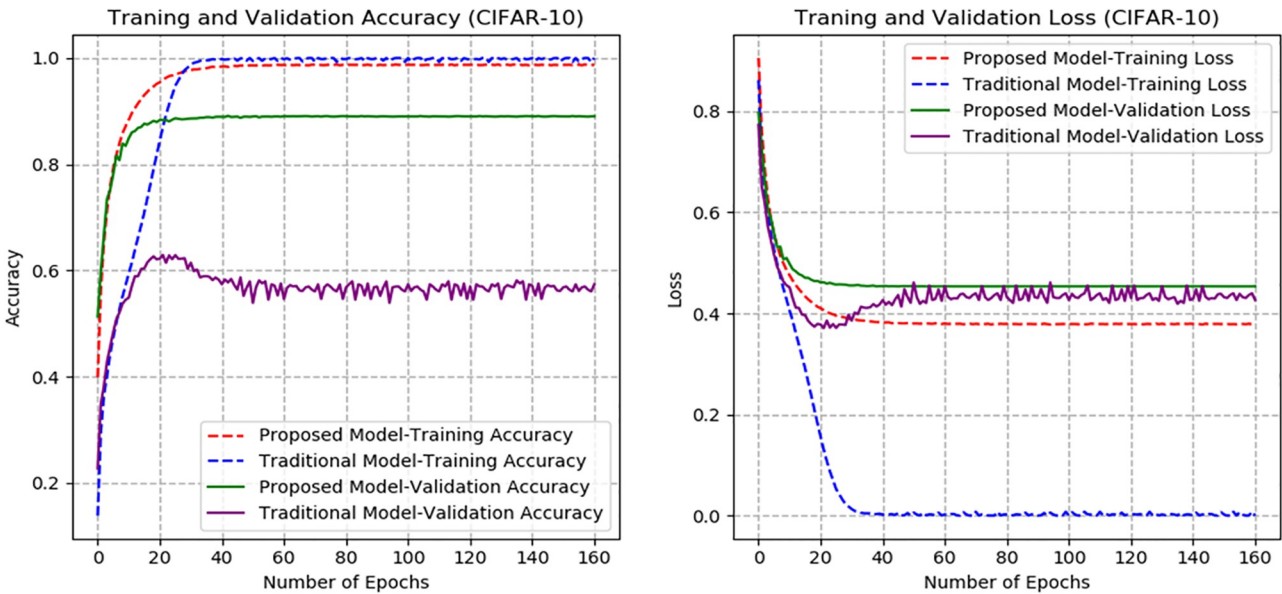

**Fig 8. Accuracy and Loss graphs of the proposed and baseline CapsNet models (CIFAR-10).**

true positive, false positive, true negative, false negative, accuracy per class values, specificity, sensitivity, and precision to extract relevant information from these figures. Considering Figs 11 and 12, Tables 1 and 2, it can be seen that the proposed model classifies images belonging to the various classes for the Brain Tumor and Breast Cancer datasets better, with high validation accuracies per class. It can be seen that; the proposed model had less images predicted that does not belong to their respective classes, hence the high values of precisions attained by the

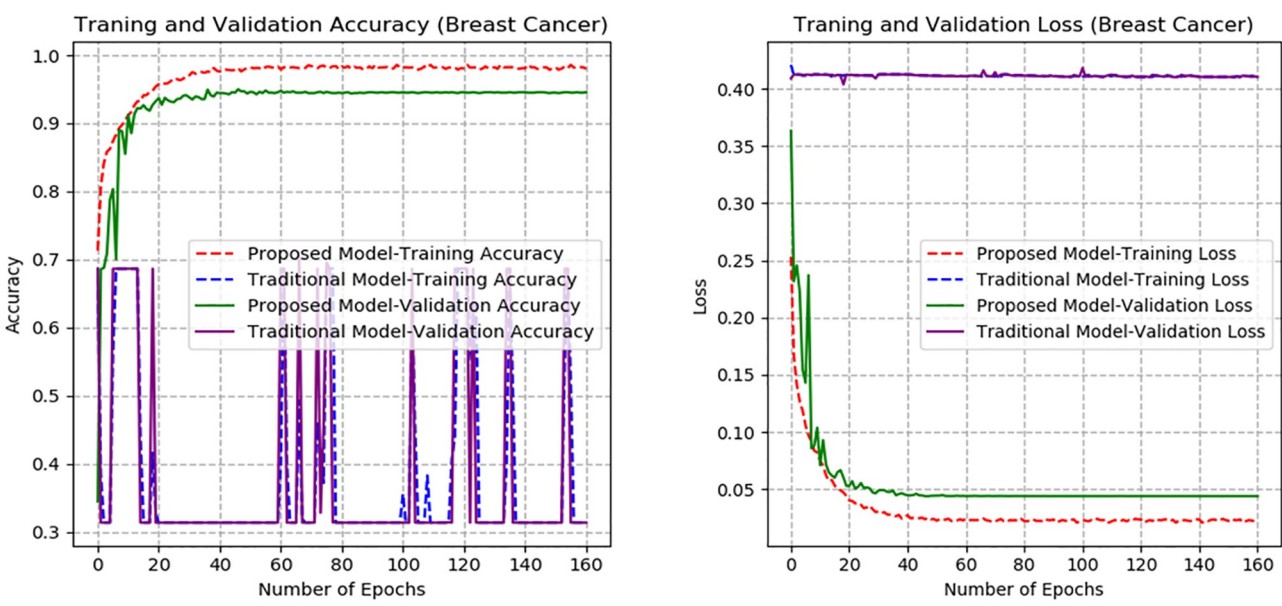

**Fig 9. Accuracy and Loss graphs of the proposed and baseline CapsNet models (Breast Cancer).**

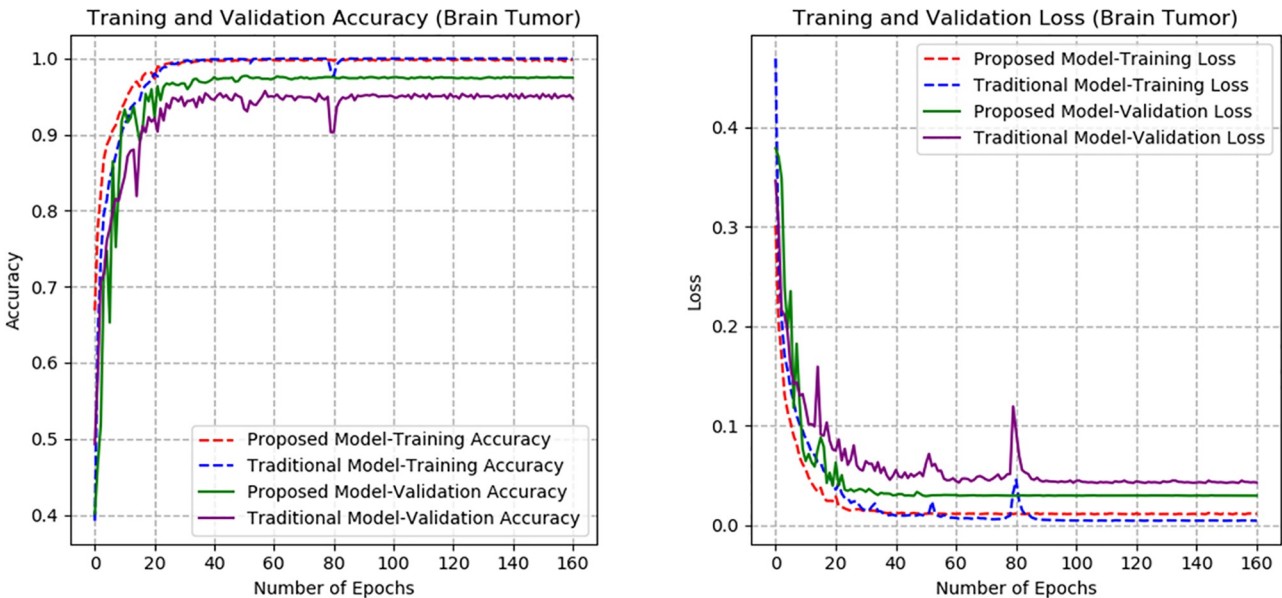

**Fig 10. Accuracy and Loss graphs of the proposed and baseline CapsNet models (Brain Tumor).**

various classes in the datasets when using the proposed model. Also, the proposed model did not miss much images belonging to the various classes, but correctly identified images belonging to particular classes, hence the higher values obtained for sensitivity or recall for the various classes in the datasets. Again, the proposed model was able to exclude images that does not

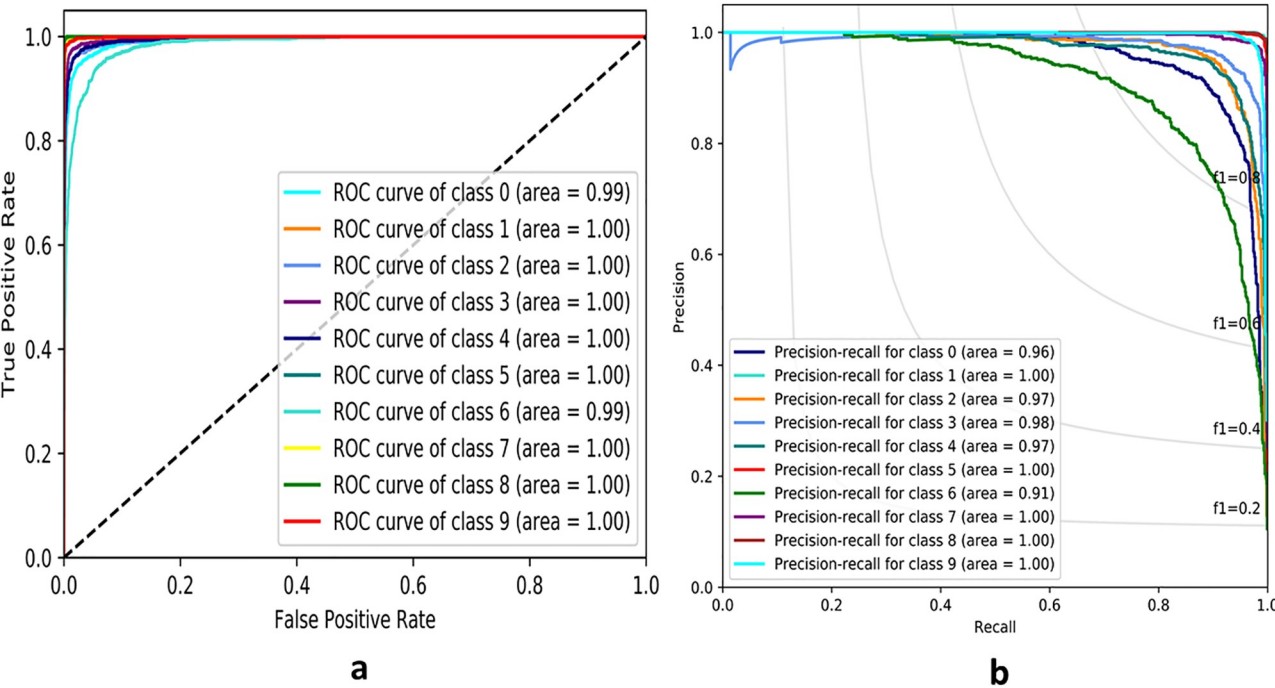

**Fig 11. Confusion Matrices of Breast Cancer: (a)** confusion matrix for the proposed model **(b)** confusion matrix for the baseline CapsNet model.

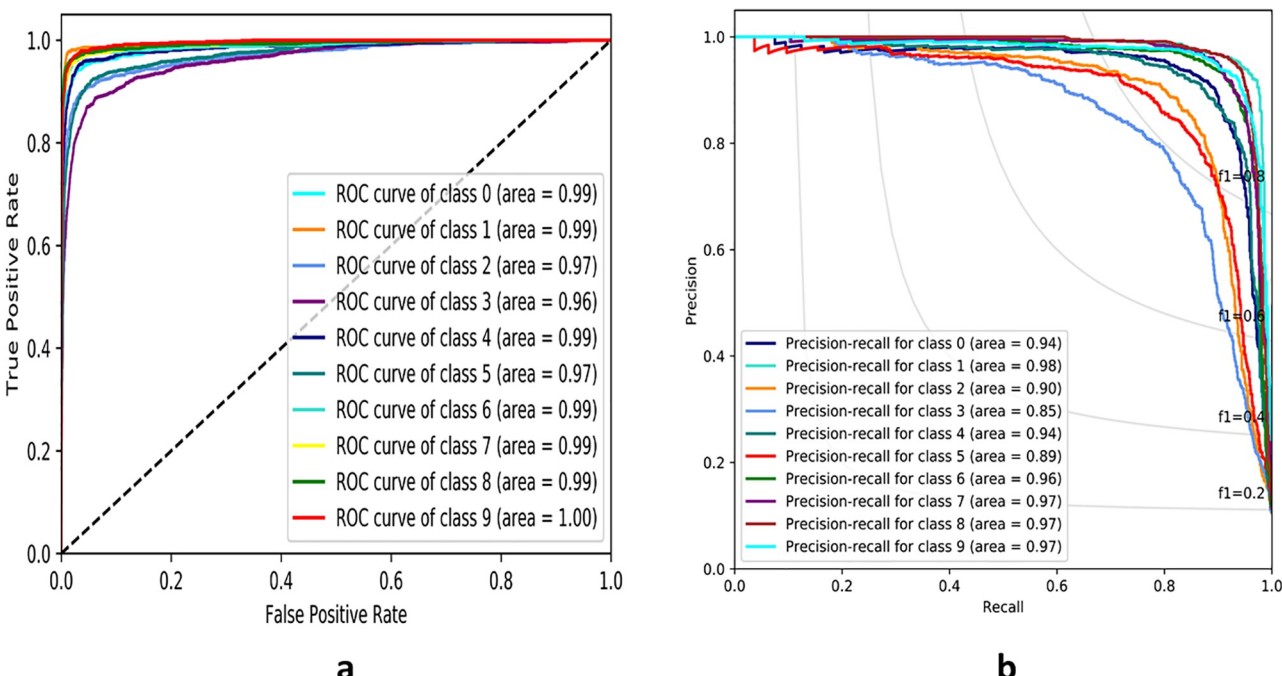

**Fig 12.** Confusion Matrices of Brain Tumor: **(a)** confusion matrix for the proposed model **(b)** confusion matrix for the baseline CapsNet model.

**Table 1. Performance metrics on the Breast Cancer (BC) dataset for the proposed model and the baseline CapsNet model.**

| Model(Dataset) | Class | TP | FP | TN | FN | Precision | Sensitivity | Specificity | Accuracy | Data Size |
|---|---|---|---|---|---|---|---|---|---|---|
| Baseline(BC) | 0 | 63 | 24 | 1062 | 433 | 0.7241 | 0.1270 | 0.9779 | 71.11% | 496 |
| | 1 | 1062 | 433 | 63 | 24 | 0.7104 | 0.9779 | 0.1270 | 71.11% | 1086 |
| Proposed(BC) | 0 | 461 | 44 | 1042 | 35 | 0.9129 | 0.9294 | 0.9595 | 95.01% | 496 |
| | 1 | 1042 | 35 | 461 | 44 | 0.9675 | 0.9595 | 0.9294 | 95.01% | 1086 |

belong to a particular class in the datasets. This is indicated by the high specificity values obtained by the various classes in the dataset when the proposed model is used. Moreover, from Figs 13 and 14, the Area under the curve (AUC) graphs for fashion-MNIST and CIFAR 10 are shown respectively. The proposed model shows better ROC values for all the classes in

**Table 2. Performance metrics on the Brain Tumor (BT) dataset for the proposed model and the baseline CapsNet model.**

| Model(Dataset) | Class | TP | FP | TN | FN | Precision | Sensitivity | Specificity | Accuracy | Data Size |
|---|---|---|---|---|---|---|---|---|---|---|
| Baseline(BT) | 0 | 271 | 16 | 995 | 29 | 0.9443 | 0.9033 | 0.9842 | 96.57% | 300 |
| | 1 | 283 | 30 | 975 | 23 | 0.9042 | 0.9248 | 0.9702 | 95.96% | 306 |
| | 2 | 296 | 6 | 1005 | 4 | 0.9801 | 0.9867 | 0.9941 | 99.24% | 300 |
| | 3 | 405 | 4 | 902 | 0 | 0.9902 | 1 | 0.9956 | 99.70% | 405 |
| Proposed(BT) | 0 | 287 | 6 | 1005 | 13 | 0.9795 | 0.9567 | 0.9941 | 98.55% | 300 |
| | 1 | 292 | 9 | 996 | 14 | 0.9701 | 0.9543 | 0.9911 | 98.25% | 306 |
| | 2 | 297 | 9 | 1002 | 3 | 0.9706 | 0.990 | 0.9911 | 99.09% | 300 |
| | 3 | 405 | 6 | 900 | 0 | 0.9854 | 1 | 0.9934 | 99.54% | 405 |

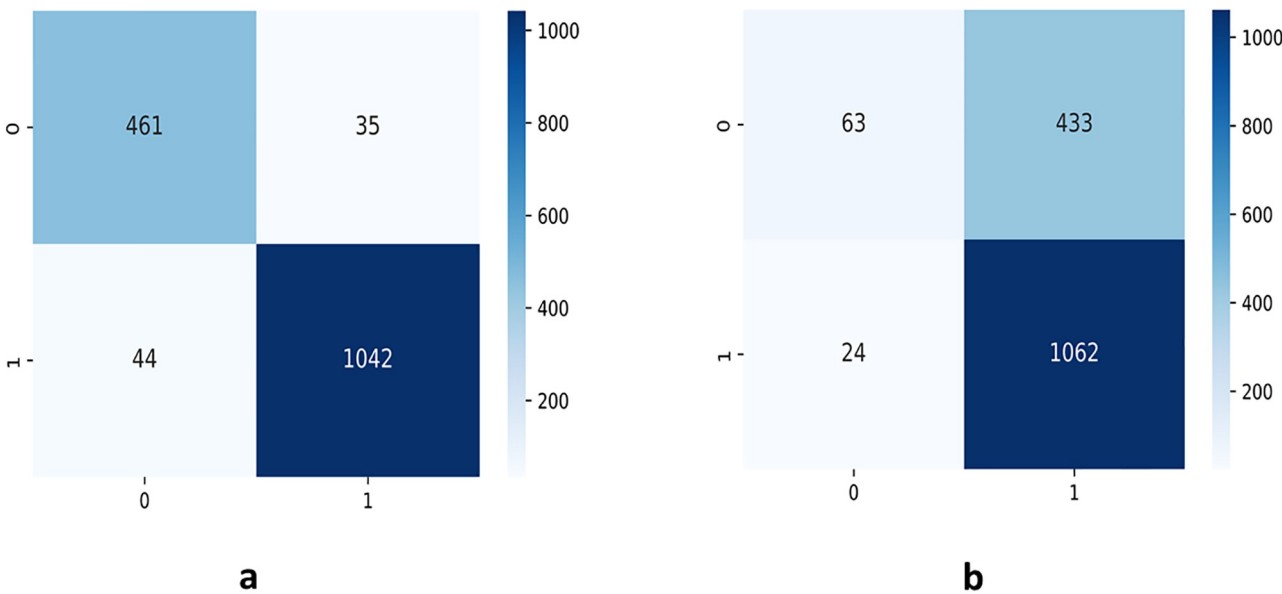

**Fig 13. Receiver Operating Characteristics and Precision Recall Curves of fashion-MNIST for the proposed model. (a)** ROC Curve **(b)** PR Curve

the datasets, which indicates that the proposed model is capable of distinguishing between classes at various threshold. Also, the precision recall values (that help to access the performance of a model on imbalanced datasets as can be found in the Breast Cancer dataset) for the proposed model were high for all the datasets. This shows that, proposed model is robust and performs well on imbalanced datasets as can be found in the breast cancer datasets. In general, the proposed model demonstrated highly encouraging outcomes in accuracy, reliability, and

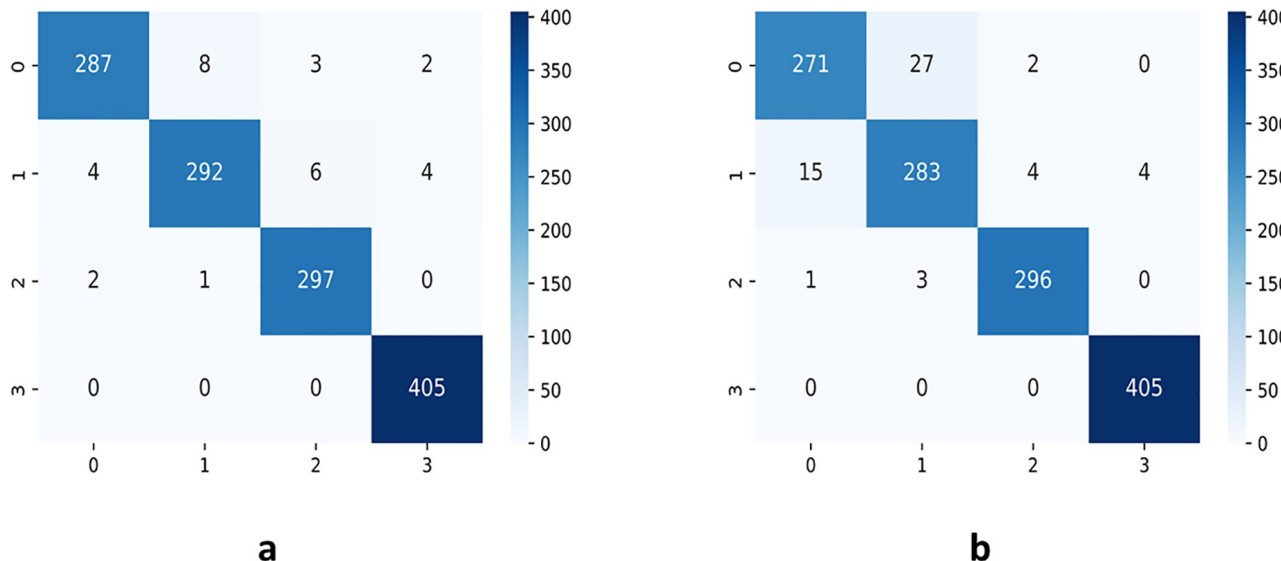

**Fig 14. Receiver Operating Characteristics (ROC) and Precision Recall Curves (PR) of CIFAR-10 for the proposed model. (a)** ROC Curve **(b)** PR Curve.

likelihood of false positive and negative predictions, which are very needed characteristics in medical image diagnosis.

The proposed model's performance is also assessed with that of the traditional CapsNet [28]. The accuracy and loss graphs for the proposed and baseline CapsNet models are displayed in Figs 7–10 for fashion-MNIST, CIFAR-10, Breast Cancer, and Brain Tumor respectively. It is clear from the accuracy graphs that the proposed model outperforms the baseline CapsNet model in terms of accuracy and early convergence. It can be seen from the accuracy and loss graphs that, the validation accuracies of the proposed model are higher for all the datasets than the baseline model. This means that, the proposed model obtained higher correct predictions from the entire number of predictions, hence, generalizes well on unseen data than the baseline model. Considering Fig 9, representing accuracy and loss for Breast Cancer dataset, it can be seen that, the baseline model underfitted and experienced lower validation accuracy with significant swings in contrast to the proposed model's graphs that experienced early convergence and higher accuracy. This means that, the baseline model excelled at fitting the training data for Breast Cancer dataset, but struggled to generalize to new or unseen Breast Cancer data during validation. Also, contrary to the proposed model, the baseline CapsNet model for CIFAR-10 exhibits a steady fall in test accuracy following convergence.

Figs 13 and 14 displays the ROC and Precision Recall curves for fashion-MNIST and CIFAR-10 respectively. The proposed model outperformed the baseline model when the values for the ROC and Precision-Recall curves were taken into account. This shows that, the proposed model is more robust.

The confusion matrices, which are depicted in Figs 11 and 12, indicates how many images were correctly and incorrectly identified by the proposed and baseline models. The diagonal cells highlighted in blue represent accurate predictions made by the models, including both true positives and true negatives. Misclassifications produced by the model are shown in white, both above and below the correct predictions. We calculated the true positive, false positive, true negative, false negative, accuracy per class values, specificity, sensitivity, and precision to extract relevant information from these figures as shown in Tables 1 and 2. In the proposed model, fewer images were incorrectly identified than in the baseline CapsNet model. It is clear from the true positive and accuracy numbers for each class that the suggested model performed well across all of the classes.

For the accuracy per class using the breast cancer dataset, for both classes 0 and 1 using the baseline model, 28.89% of the images belonging to these classes were wrongly classified as compared to the proposed model that only misclassified 4.99% of the images belonging to these classes. For the brain tumor dataset, misclassification for classes 0, 1, 2, and 3 using the baseline model was 3.43%, 4.04%, 0.76%, and 0.3% respectively, whiles the proposed model misclassifications were 1.45%, 1.75%, 0.91%, and 0.45% respectively for the same classes. This shows that, the proposed model classifies images belonging to the various classes better than the baseline model. Also, for precision using the breast cancer dataset, for both classes 0 and 1 using the baseline model, 27.59% and 28.96% of images predicted to belong to classes 0 and 1 respectively do not belong to those classes, compared to the proposed model, where only 8.71% and 3.2% images for classes 0 and 1 respectively do not belong to those classes. For the brain tumor, using the baseline model, 5.57%, 9.58%, 1.99%, and 0.98% of images for classes 0, 1, 2, and 3 respectively, did not actually belong to these classes compared to 2.05%, 2.99%, 2.94%, and 1.46% of images for classes 0, 1, 2, and 3 respectively, when using the proposed model. It can be seen that, the proposed model had fewer images predicted that do not belong to their respective classes than the baseline model. Again, for sensitivity using the breast cancer dataset, for both classes 0 and 1 with the baseline model, 87.3% and 2.21% images belonging to these classes are missed, compared to 7.06% and 4.05% of images missed for classes 0 and 1

respectively using the proposed model. For the brain tumor dataset, using the baseline model, 9.67%, 7.52%, 1.33%, and 0% of images for classes 0, 1, 2, and 3 respectively are missed as compared to 4.33%, 4.57%, 1%, and 0% of images for classes 0, 1, 2, and 3 respectively, missed using the proposed model. This shows that, the proposed model did not miss many images belonging to the various classes, but correctly identified images belonging to particular classes as opposed to the baseline model. Also, for specificity using the breast cancer dataset, for both classes 0 and 1 with the baseline model, 2.21% and 87.3% of images were misclassified as belonging to the various classes respectively, whiles the proposed model misclassified 4.05% and 7.06% images for the various classes respectively as belonging to those classes, whiles they do not belong to them. For the brain tumor, using the baseline model, 1.58%, 2.98%, 0.59%, and 0.44% of images for classes 0, 1, 2, and 3 respectively, were misclassified as belonging to the various classes, compared to 0.59%, 0.89%, 0.89%, and 0.66% images for classes 0, 1, 2, and 3 respectively, that were misclassified to belong to the various classes by the proposed model whiles they did not belong to them. These figures obtained, shows that the proposed model was able to exclude images that does not belong to a particular class better than the baseline model. All this can be found in Tables 1 and 2 whose values were retrieved from confusion matrices found in Figs 11 and 12. Furthermore, the proposed model gives better ROC values than the baseline model, which indicates that the proposed model is capable of distinguishing between classes at various thresholds. Also, the precision recall values for the proposed model were better compared to that of the baseline model as shown in Figs 13 and 14. These graphs help to access a model's robustness, when the dataset is imbalanced as can be found in the breast cancer dataset. Overall, the proposed model showed very promising results in terms of accuracy, reliability, and likelihood of false positive and negative predictions, compared to the baseline model, which is a very needed property, in medical image diagnosis. These promising results can be attributed to the good feature extraction capabilities (texton and dense layers) that were employed to extract only relevant information from the various images subjected to the model.

## Ablation study

To study a models' elements that significantly influenced its performance, ablation experiments are carried out [97]. The texton and dense layers are firstly removed and then added sequentially for the objectives of this work. On the fashion-MNIST, CIFAR-10, Breast Cancer, and Brain Tumor Datasets, it can be seen that the model achieves validation accuracies of 94.65%, 83.19%, 91.72%, and 93.71%, respectively, with only one dense layer implemented. The performance of the suggested model is significantly enhanced by the addition of the texton and dense layers. Table 3 demonstrates this.

**Table 3. Ablation study results (Validation accuracy in (%)).**

| Layers | fashion-MNIST | CIFAR-10 | Breast Cancer | Brain Tumor |
|---|---|---|---|---|
| + Dense Layer 1 | 94.65 | 83.19 | 91.72 | 94.97 |
| + Texton Layer 1 | 94.72 | 84.91 | 91.78 | 94.98 |
| + Dense Layer 2 | 94.75 | 87.76 | 94.25 | 96.87 |
| + Texton Layer 2 | 94.77 | 87.83 | 94.50 | 97.18 |
| +Dense Layer 3 | 94.80 | 88.55 | 94.69 | 94.48 |
| + Texton Layer 3 (all layers) | 94.90 | 89.09 | 95.01 | 97.71 |

## Number of parameters

On complex images, most models in literature increases the width and depth of models to perform well on complex images. The number of parameters increases as a result of this. For instance, 138, 60, 23 million parameters are generated by VGG16 [98], AlexNet [99], ResNet50 [100], etc. A model's complexity is dependent on number of parameters and this places a heavy computational burden on system resources. This limits their ability to be used on devices with less memory, such as mobile phones. The number of parameters generated for the various datasets in millions for the proposed model are 11.89, 14.53, 8.88, 10.29, and that for the baseline model are 8.22, 11.75, 4.93, and 9.43 for fashion-MNIST, CIFAR-10, Breast Cancer and Brain Tumor datasets respectively. It can be seen that; the proposed model generates more parameters and hence spends more time in training than the baseline model, but the proposed model is more optimal in terms of precision, sensitivity, specificity, accuracy, ROC, and PR curve values.

## Comparison of results

Here comparison of the performance of the proposed model to the state-of-the-art models applied to CIFAR-10 and fashion-MNIST in literature in order to show how well it performs is done. The architectural level is where we made the change. Despite the fact that our work focuses on dynamic routing, we broaden our comparison to include several routing techniques.

As shown in Table 4, apart from DenseCaps that achieved slightly higher values on fashion-MNIST and CIFAR-10, considering the fashion-MNIST dataset, the proposed model that employed texton and dense layers for better feature extraction obtained the highest validation accuracy of 94.90%, followed by Abra and colleagues' proposed CapsNet model [45] that used Gabor and preprocessing blocks for feature extraction and the least validation accuracy of 90.26% was obtained from Ozan and colleagues proposed model [43] that used quaternions to design a CapsNet model. For CIFAR-10 dataset, a CapsNet model by Phaye and colleagues [41], that used dense blocks for feature extraction achieved the highest validation accuracy of 89.71, followed by the proposed model that achieved a validation accuracy of 89.09%, and the least validation accuracy is from the baseline CapsNet by Sabour and colleagues [28], which uses a convolution layer for feature extraction. For the breast cancer dataset, the proposed model had the highest validation accuracy of 95.01% as compared to 71.11% of the basline model. Lastly, for the Brain tumor dataset, the proposed model had the highest validation accuracy of 97.71% followed by the baseline model at 95.73%, and then the model proposed by Phaye and colleagues obtained the least validation accuracy of 95.03% [41]. The proposed model performs commensurately well compared to the various state-of-the-art capsule network models. The proposed model outperformed the baseline model by 4.18%, 26.18% 23.9% and, 1.98% for fashion-MNIST, CIFAR10, Breast Cancer, and Brain Tumor datasets.

This desirable performance required for complex image (e.g., medical image) diagnosis by the proposed model is attributed to the good feature extraction capabilities (texton and dense layers) that were employed to extract only relevant information from the various images subjected to the model.

## Transformed data learning and transformation robustness

The purpose and impact of transformations applied to the training and testing datasets serve distinct objectives. Transformations during testing can serve as a means to assess a model's robustness to transformations. Conversely, transformations during training serve as a way to evaluate a model's capacity to learn from these transformations. Both aspects contribute to

**Table 4. Proposed model and previous works comparison (Validation accuracy in (%)).** Unreported values are indicated with (-).

| Methods | fashion-MNIST | CIFAR-10 | Breast Cancer | Brain Tumor |
|---|---|---|---|---|
| ShallowNet [33] | 92.70 | 75.75 | - | - |
| CapsNet [28] | 90.72 | 62.91 | 71.11 | 95.73 |
| Enhanced-CapsNet [34] | - | 82.31 | - | - |
| 64 Capsule Layers [31] | - | 64.67 | - | - |
| Feature Amplification CapsNet [35] | 93.76 | 84.56 | - | - |
| Multi-lane [32] | 92.63 | 76.79 | - | - |
| MS-CapsNet [36] | 92.70 | 75.70 | - | - |
| ResCapsNet [37] | - | 78.54 | - | - |
| CFC-CapsNet [38] | 92.86 | 73.15 | - | - |
| Fast Inference [39] | 91.52 | 70.33 | - | - |
| Inverted dot product [40] | - | 82.55 | - | - |
| DCNET++ [41] | 94.65 | 89.71 | - | 95.03 |
| Max–min [42] | 92.07 | 75.92 | - | - |
| Quaternion CapsNet [43] | 90.26 | 82.21 | - | - |
| Gabor capsNet with prep blocks [45] | 94.78 | 85.24 | - | - |
| MLSCN [46] | - | 76.79 | - | - |
| DenseCaps [47] | 94.93 | 89.41 | - | - |
| DeeperCaps [48] | - | 81.29 | - | - |
| MLCN [49] | 92.63 | 75.18 | - | - |
| Cv-CapsNet++ [50] | 94.40 | 86.70 | - | - |
| Quick-CapsNet (QCN) [51] | 88.84 | 67.18 | - | - |
| DA-CapsNet [52] | 93.98 | 85.47 | - | - |
| R-CapsNet [53] | 93.89 | 81.57 | - | - |
| SqueezeCapsNet [101] | 93.49 | 82.45 | - | - |
| Afriyie et al [102] | 92.80 | 75.42 | - | - |
| MCNet [103] | 93.17 | 79.27 | - | - |
| Afriyie et al [104] | 94.93 | 84.57 | - | - |
| **Proposed Model** | **94.90** | **89.09** | **95.01** | **97.71** |

understanding a model's transformation robustness and the learning ability from data that is transformed. In this research, we introduced random transformations to the train set of the fashion-MNIST and CIFAR-10 datasets, including translations of up to 2 pixels in both the horizontal and vertical directions and random rotations within the (-180°, 180°) range. This is denoted as a tuple (2, 180°). To assess model robustness, we created a test set with translational shifts of 2 pixels and rotational variations within 90°, represented as a tuple (2, 90°). To ensure a comprehensive evaluation of the proposed model's robustness, we included two CNN models, ResNet-18-aug and ResNet-34-aug, which were trained with augmentation techniques, alongside the baseline CapsNet model. The performance of the two CNN baseline models on fashion-MNIST and CIFAR-10 datasets has been documented in [105], and these results serve as points of reference for comparison in Table [table number]. From Table 5 it can be seen that, even though there are small reductions of 0.16% and 0.23% in the validation accuracies 94.90% and 89.09% obtained for fashion-MNIST and CIFAR-10 transformed image datasets respectively for the proposed model. The proposed model generalized very well on the transformed unseen data, hence achieving the highest validation accuracies on the two datasets. This shows the proposed models' robustness to image transformation. Also, the three baseline models also performed comparatively well in terms of robustness to image transformation.

**Table 5. Comparison of validation accuracies in % for transformation robustness of CapsNet and CNN based models.**

|  | fashion-MNIST | fashion-MNIST | CIFAR-10 | CIFAR-10 |
|---|---|---|---|---|
| Methods | (0, 0) | (2, 90˚) | (0, 0) | (2, 90˚) |
| ResNet-18-aug [105] | 94.21 | 93.30. | 78.84 | 79.60 |
| ResNet-34-aug [105] | 94.38 | 93.78 | 81.27 | 81.60 |
| CapsNet [28] | 90.72 | 88.52 | 62.91 | 61.46 |
| Proposed Model | 94.90 | 94.74 | 89.09 | 88.86 |

There was a reduction in the validation accuracies 94.21% and 94.38% by 0.91% and 0.60% respectively, when using fashion-MNIST transformed image dataset for the two CNN baseline models ResNet-18-aug and ResNet-34-aug respectively, but an increase in the validation accuracies 78.84% and 81.27% by 0.76% and 0.33% respectively when using the same CNN baseline models on CIFAR-10 transformed image dataset. Again, the baseline CapsNet model when used, recorded a reduction in validation accuracy by 1.45% and 2.2% when fashion-MNIST and CIFAR-10 transformed image datasets were run with the model respectively. Comparing the proposed model to the baseline CNN and CapsNet models, it can clearly be seen that, the proposed model is more robust to image transformation. This splendid performance can be attributed to the equivariant nature of CapsNet, as well as the effective feature extraction capabilities of the proposed model.

## Conclusion and future work

An improved Capsule Network model with a texton detection method and dense layers was proposed in this paper to help extract key features and increase the recognition rate on complex images. The adoption of texton detection method and dense layers helped to improve the model's feature extraction abilities, thus helping to prevent overfitting due to class imbalance, acquiring competitive validation accuracies, and obtaining performances that are appreciable. The proposed model outperformed the CapsNet baseline model and performed comparatively well with the state-of-the-art models in literature on the four datasets in terms of convergence, accuracy, flexibility, robustness, and complexity. When tested against the fashion-MNIST, CIFAR-10, breast cancer, and brain tumor datasets, the suggested model performed admirably. In general, the results of this study clarify the viability of using Capsule Networks on complex tasks in the real world. However, the routing algorithm used by the capsule network has been blamed for decreasing its expressivity as well as hindering it from input distinguishing, which limits the effective classification of the model. In the future, a new routing algorithm coupled with fewer model parameters will be considered to improve the routing of vectors from lower-level to higher-level capsules for better classification and feasible implementation on smartphones.

## Author Contributions

**Conceptualization:** Vivian Akoto-Adjepong.

**Data curation:** Vivian Akoto-Adjepong.

**Formal analysis:** Vivian Akoto-Adjepong.

**Investigation:** Vivian Akoto-Adjepong.

**Methodology:** Vivian Akoto-Adjepong.

**Project administration:** Obed Appiah, Patrick Kwabena Mensah, Peter Appiahene.

**Resources:** Vivian Akoto-Adjepong.

**Software:** Vivian Akoto-Adjepong.

**Supervision:** Obed Appiah, Patrick Kwabena Mensah, Peter Appiahene.

**Validation:** Vivian Akoto-Adjepong.

**Visualization:** Vivian Akoto-Adjepong.

**Writing – original draft:** Vivian Akoto-Adjepong.

**Writing – review & editing:** Obed Appiah, Patrick Kwabena Mensah, Peter Appiahene.

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
