## [Decision Letter · Decision Letter 0]

18 Apr 2023

PONE-D-23-06111TTDCapsNet: Tri Texton-Dense Capsule Network for Complex image recognitionPLOS ONE

Dear Dr. Akoto-Adjepong,

Thank you for submitting your manuscript to PLOS ONE. After careful consideration, we feel that it has merit but does not fully meet PLOS ONE’s publication criteria as it currently stands. Therefore, we invite you to submit a revised version of the manuscript that addresses the points raised during the review process.

Please address the major concerns in the introduction and conclusion section. Without these changes, the paper cannot be accepted. If reviewers have mentioned new articles for citation, only add these articles if they are relevant to your research.

We look forward to receiving your revised manuscript.

Kind regards,

Rahul Gomes, Ph.D.

Academic Editor

PLOS ONE

Journal Requirements:

Reviewers' comments:

Reviewer's Responses to Questions

**Comments to the Author**

1. Is the manuscript technically sound, and do the data support the conclusions?

Reviewer #1: Partly

Reviewer #2: Yes

2. Has the statistical analysis been performed appropriately and rigorously? 

Reviewer #1: I Don't Know

Reviewer #2: Yes

3. Have the authors made all data underlying the findings in their manuscript fully available?

Reviewer #1: No

Reviewer #2: Yes

4. Is the manuscript presented in an intelligible fashion and written in standard English?

Reviewer #1: Yes

Reviewer #2: Yes

5. Review Comments to the Author

Reviewer #1: 1. Points in favor

i. The article contributes to knowledge and may be considered for publication after carrying out the following suggestions to improve the paper

ii. Overall, the article is ok

2. For the authors

i. The abstract needs to be rewritten in one paragraph stating clearly what is done, how its done and results/limitations

ii. The introduction is not properly written. A good introduction should lead the reader from a generalized topic to a particular aspect. It helps to establish the main idea, context, and research importance and summarizes background data on the topic, providing the primary goal of the work. Also, the contribution of the paper is not clearly stated.

iii. The literature review needs to be improved, and the presentation style should be flow of thought. Consider the following to improve the body of literature

1. DOI: 10.1371/journal.pone.0282812

2. DOI: 10.1007/s42235-022-00316-8

3. DOI: 10.33003/fjs-2022-0603-990

4. DOI: 10.1007/s11831-022-09850-4

5. DOI: 10.1371/journal.pone.0275346

6. DOI: 10.1007/s00521-022-07854-6

iv. All Figures need to be explained in more detail as it relates to the study.

v. The result and discussion section are poorly written. The authors need to explain or link the finding with their proposed contributions and how it differs from what is in the available literature

vi. There may be a need to find the statistical significance of the claimed improvement of this work

vii. The conclusion may need to be improved, relating it to the contributions and findings. Also, state some limitations

Reviewer #2: - Spell out each acronym the first time used in the Abstract (CNN) as well as the body of the paper

-Author should discuss more recent references in introducation as below.

*Gaze Prediction Based on Convolutional Neural Network

*Radiologists versus Deep Convolutional Neural Networks: A Comparative Study for Diagnosing COVID-19

*Accurate and compact convolutional neural network based on stochastic computing

*Automated Diagnosis of Acne and Rosacea using Convolution Neural Networks

*Deterministic Modeling to Predict the Natural Gas Density Using Artificial Neural Networks

-author should improve figure 1 and 2.background of all images should be white.

-Authors should remove ttitle above the figure 7 and 8. if they same as the caption otherwise it can be written as caption.and also background

-graphs of figure 7 and 8 are not looking good.author should improve the quality.

-Conclusion to be made more systematic and future scope to be elaborated more on technical features that are planned to be added in the proposed system in the near future.

- In Conclusion, future directions and challenges should be explained more.

- please give a proofread check to the paper.

6. PLOS authors have the option to publish the peer review history of their article (what does this mean?). If published, this will include your full peer review and any attached files.

Reviewer #1: No

Reviewer #2: No

---

## [Author Response · Author response to Decision Letter 0]

2 Jun 2023

Original Manuscript ID: PONE-D-23-06111

Original Article Title: “TTDCapsNet: Tri Texton-Dense Capsule Network for Complex image recognition”

To: PLOS ONE

Re: Response to reviewers

Dear Editor,

We would like to express our profound gratitude to the Editor and the reviewers for their professional and constructive comments on our manuscript, “TTDCapsNet: Tri Texton-Dense Capsule Network for Complex image Recognition” (ID: PONE-D-23-06111). We have revised the manuscript according to the comments and suggestions of the associate Editor and the reviewers, and have responded to the reviewer’s concerns. The responses to the reviewer’s comments are therefore attached in separate sheets.

Thank you for allowing a resubmission of our manuscript, with an opportunity to address the reviewers’ comments.

We are uploading (a) our point-by-point response to the comments (below) (response to reviewers), (b) an updated manuscript with yellow and green highlighting indicating changes for made as suggested by Reviewer1 and 2 respectively (some of the changes to be made overlap), and (c) a clean, updated manuscript without highlights (PDF main document). Finally, the authors would like to affirm your office that the manuscript can be assigned to the same reviewers. For further information, please feel free to contact the corresponding author. Her personal information is as follows:

Mailing address: Ms. Vivian Akoto-Adjepong 

Department of Computer Science and Informatics, 

University of Energy and Natural Resources, 

P.O. Box 214, Sunyani - Ghana. 

Phone number: +233 (0) 247385738

Email: vivian.akoto-adjepong@uenr.edu.gh

Best regards

Vivian Akoto-Adjepong et al.

Reviewer #1: 1. Points in favor

i. The article contributes to knowledge and may be considered for publication after carrying out the following suggestions to improve the paper

Response: Thank you for the review

ii. Overall, the article is ok

Response: Thank you for the review

Reviewer 1, Comment # 1: The abstract needs to be rewritten in one paragraph stating clearly what is done, how its done and results/limitations

Response: Thank you for the review. In the revised manuscript, the abstract has been rewritten in one paragraph stating clearly what is done, how it’s done, and the results. [see page: 1]

Author action: 

Convolutional Neural Networks (CNNs) are frequently used algorithms because of their propensity to learn relevant and hierarchical features through their feature extraction technique. However, the availability of enormous volumes of data in various variations is crucial for their performance. Capsule networks (CapsNets) perform well on a small amount of data but perform poorly on complex images. To address this, we proposed a new Capsule Network architecture called Tri Texton-Dense CapsNet (TTDCapsNet) for better complex image classification. The TTDCapsNet is made up of three hierarchic blocks of Texton-Dense CapsNet (TDCapsNet) models. A single TDCapsNet is a CapsNet architecture composed of a texton detection layer to extract essential features, which are passed onto an eight-layered block of dense convolution that further extracts features, and then the output feature map is given as input to a Primary Capsule (PC), and then to a Class Capsule (CC) layer for classification. The resulting feature map from the first PC serves as input into the second-level TDCapsNet, and that from the second PC serves as input into the third-level TDCapsNet. The routing algorithm receives feature maps from each PC for the various CCs. Routing the concatenation of the three PCs creates an additional CC layer. All these four feature maps combined, help to achieve better classification. On fashion-MNIST, CIFAR-10, Breast Cancer, and Brain Tumor datasets, the proposed model is evaluated and achieved validation accuracies of 94.90\\%, 89.09\\%, 95.01\\%, and 97.71\\% respectively. Findings from this work indicate that TTDCapsNet outperforms the baseline and performs comparatively well with the state-of-the-art CapsNet models using different performance metrics. This work clarifies the viability of using Capsule Network on complex tasks in the real world. Thus, the proposed model can be used as an intelligent system, to help oncologists in diagnosing cancerous diseases and administering treatment required. 

Reviewer 1, Comment # 2: The introduction is not properly written. A good introduction should lead the reader from a generalized topic to a particular aspect. It helps to establish the main idea, context, and research importance and summarizes background data on the topic, providing the primary goal of the work. Also, the contribution of the paper is not clearly stated.

Response: Thank you for the review. In the revised manuscript, the introduction has been written properly to improve the readability of the work, and the contribution of the paper is also clearly stated. Find some portions of the introduction below. [see page:2- 3]

Author action:

Deep learning models [10, 11] have been applied to many domains including medical health, to aid specialists, and make the classification of such diseases (malignant) much easier. They must increase performance on complicated images in terms of convergence, accuracy, flexibility, robustness, and complexity before it can be helpful to oncologists in carrying out the procedure of malignant illness identification effectively. CNNs are well-known and frequently used deep learning models because of their propensity to learn relevant and hierarchical features through their feature extraction technique with convolution structures [12]. They have been applied in the medical field, in diagnosing diseases such as Acne and Rosacea [13–15], brain tumors [16–18], breast cancer [19–21], etc. [see page: 2]

The idea of ” capsules” was first proposed to address the issues of CNN [23, 25] [see page: 2]

Therefore, there is a need to improve the existing capsule network algorithm to help solve the problem of crowding, in order to classify complex images well or with precision. This paper utilizes the dynamic routing algorithm of Capsule Network by incorporating a texton layer [30] to extract essential features, which are passed onto an eight (8)-layered block of dense convolution that further extracts features to improve the texture, color, and spatial recognition capabilities of Capsule Network, by enabling the decision of crucial features, and the needed coupling coefficients to be decreased in order to improve the hierarchical relationship among capsules that are related closely for better classification of complex images such as medical images. Fig 1 depicts the workflow adopted for the proposed work, which depicts the testing and training processes. [see page: 2-3]

This paper’s main contributions are as follows:

1) An improved, flexible and robust Capsule Network which incorporates texton and eight (8)-layered blocks of dense convolution for effective feature extraction is proposed. 

2) Excellent color, edge, and texture extraction algorithms texton and dense convolution were suggested for better feature extraction, to help in identifying affected parts of medical images using Capsule Network.

3) A comparative analysis was conducted to evaluate the proposed model with other deep learning models, using metrics such as specificity, sensitivity, precision, accuracy, and others. [see page: 3]

Reviewer 1, Comment # 3: The literature review needs to be improved, and the presentation style should be flow of thought. Consider the following to improve the body of literature

1. DOI: 10.1371/journal.pone.0282812

2. DOI: 10.1007/s42235-022-00316-8

3. DOI: 10.33003/fjs-2022-0603-990

4. DOI: 10.1007/s11831-022-09850-4

5. DOI: 10.1371/journal.pone.0275346

6. DOI: 10.1007/s00521-022-07854-6

Response: Thank you for the review. In the revised manuscript, the authors have improved the literature review and the presentation style. [see page: 3] [see page:5]

Author action:

This review starts with capsule networks proposed for the recognition of open datasets for deep learning models, CIFAR-10, Fashion MNIST, etc., and other datasets. [see page: 3]

Other authors proposed CapsNet and other deep-learning models in order to identify malignant conditions for breast cancer and brain tumor detection. [see page: 5]

All the proposed models performed well on the various datasets. But for medical image diagnosis, there is a need for a more robust and efficient model for better diagnosis, hence this study aims to propose an improved, flexible, and robust Capsule Network which incorporates texton and eight (8)-layered block of dense convolution for effective feature extraction, for better classification of malignant diseases. [see page: 8]

Reviewer 1, Comment # 4: All Figures need to be explained in more detail as it relates to the study.

Response: Thank you for the review. In the revised manuscript, all figures have been explained in detail as it relates to the study. Here are some of the explanations. [see page: 14, 15]

Author action:

The proposed model’s performance is assessed with that of the traditional CapsNet [28]. The accuracy and loss graphs for the proposed and baseline CapsNet models are displayed in Fig 7. It is clear from the accuracy graphs that the proposed model outperforms the baseline CapsNet model in terms of accuracy and early convergence. This means that the proposed model predicts correctly more than the baseline model. In comparison to the proposed model validation accuracy, which is greater and consistent, the baseline validation accuracy for breast cancer experienced lower accuracy with significant swings. Also, contrary to the proposed model, the baseline CapsNet model for CIFAR-10 exhibits a steady fall in test accuracy following convergence. 

Fig 8 displays the ROC and area under the curve graphs for fashion-MNIST, CIFAR-10, Breast Cancer, and Brain Tumor. The proposed model outperformed the baseline model when the values for the ROC and Precision-Recall curves were taken into account. This shows that, the proposed model is more robust.

The confusion matrices, which are depicted in Fig 9, indicate how many images were correctly and incorrectly identified. We calculated the true positive, false positive, true negative, false negative, accuracy per class values, specificity, sensitivity, and precision to extract relevant information from these figures. In the proposed model, fewer images were incorrectly identified than in the baseline CapsNet model, as shown in Fig 9. It is clear from the true positive and accuracy numbers for each class that the suggested model performed well across all of the classes. [see page: 14]

For the accuracy per class using the breast cancer dataset, for both classes 0 and 1 using the baseline model, 28.89% of the images belonging to these classes were wrongly classified as compared to the proposed model that only misclassified 4.99% of the images belonging to these classes. For the brain tumor dataset, misclassification for classes 0, 1, 2, and 3 using the baseline model was 3.43%, 4.04%, 0.76%, and 0.3% respectively, whiles the proposed model misclassifications were 1.45%, 1.75%, 0.91%, and 0.45% respectively for the same classes. This shows that the proposed model classifies images belonging to the various classes better than the baseline model. Also, for precision using the breast cancer dataset, for both classes 0 and 1 using the baseline model, 27.59% and 28.96% of images predicted to belong to classes 0 and 1 respectively do not belong to those classes, compared to the proposed model, where only 8.71% and 3.2% images for classes 0 and 1 respectively does not belong to those classes. For the brain tumor, using the baseline model, 5.57%, 9.58%, 1.99%, and 0.98% of images for classes 0, 1, 2, and 3 respectively, did not actually belong to these classes compared to 2.05%, 2.99%, 2.94%, and 1.46% of images for classes 0, 1, 2, and 3 respectively, when using the proposed model. It can be seen that the proposed model had fewer images predicted that do not belong to their respective classes than the baseline model. Again, for sensitivity using the breast cancer dataset, for both classes 0 and 1 with the baseline model, 87.3% and 2.21% images belonging to these classes are missed, compared to 7.06% and 4.05% of images missed for classes 0 and 1 respectively using the proposed model. For the brain tumor dataset, using the baseline model, 9.67%, 7.52%, 1.33%, and 0% of images for classes 0, 1, 2, and 3 respectively are missed as compared to 4.33%, 4.57%, 1%, and 0% of images for classes 0, 1, 2, and 3 respectively, missed using the proposed model. This shows that, the proposed model did not miss many images belonging to the various classes, but correctly identified images belonging to particular classes as opposed to the baseline model. Also, for specificity using the breast cancer dataset, for both classes 0 and 1 with the baseline model, 2.21% and 87.3% of images were misclassified as belonging to the various classes respectively, whiles the proposed model misclassified 4.05% and 7.06% images for the various classes respectively as belonging to those classes, whiles they do not belong to them. For the brain tumor, using the baseline model, 1.58%, 2.98%, 0.59%, and 0.44% of images for classes 0,1,2,3 respectively, were misclassified as belonging to the various classes, compared to 0.59%, 0.89%, 0.89%, and 0.66% images for classes 0,1,2,3 respectively, that were misclassified to belong to the various classes by the proposed model whiles they did not belong to them. These figures obtained, show that the proposed model was able to exclude images that do not belong to a particular class better than the baseline model. All this can be found in Table 1 and Table 2., whose values were retrieved from confusion matrices found in Fig 9. Furthermore, the proposed model gives better ROC values than the baseline model, which indicates that the proposed model is capable of distinguishing between classes at various thresholds. Also, the precision-recall values for the proposed model were better, compared to that of the baseline model as shown in Fig 8. These graphs help to access a model’s robustness, when the dataset is imbalanced as can be found in the breast cancer dataset. Overall, the proposed model showed very promising results in terms of accuracy, reliability, and likelihood of false positive and negative predictions, compared to the baseline model, which is a very needed property, in medical image diagnosis. These promising results can be attributed to the good feature extraction capabilities (texton and dense layers) that were employed to extract only relevant information from the various images subjected to the model. [see page: 15]

Reviewer 1, Comment # 5: The result and discussion section are poorly written. The authors need to explain or link the finding with their proposed contributions and how it differs from what is in the available literature.

Response: Thank you for the review. In the revised manuscript, the result and discussion section has been rewritten well to link the finding with the proposed contribution and how it differs from what is available in the literature. [see page: 20]

Author action:

As shown in Table 4, considering the fashion-MNIST dataset, the proposed model that employed texton and dense layers for better feature extraction obtained the highest validation accuracy of 94.90%, followed by Abra and colleagues’ proposed CapsNet model [45] that used Gabor and preprocessing blocks for feature extraction and the least validation accuracy of 90.26% was obtained from Ozan and colleagues proposed model [43] that used quaternions to design a CapsNet model. For CIFAR-10 dataset, a CapsNet model by Phaye and colleagues [41], that used dense blocks for feature extraction achieved the highest validation accuracy of 89.71, followed by the proposed model that achieved a validation accuracy of 89.09%, and the least validation accuracy is from the baseline CapsNet by Sabour and colleagues [28], which uses a convolution layer for feature extraction. For the breast cancer dataset, the proposed model had the highest validation accuracy of 95.01% as compared to 71.11% of the basline model. Lastly, for the Brain tumor dataset, the proposed model had the highest validation accuracy of 97.71% followed by the baseline model at 95.73%, and then the model proposed by Phaye and colleagues obtained the least validation accuracy of 95.03% [41]. The proposed model performs commensurately well compared to the various state-of-the-art capsule network models. The proposed model outperformed the baseline model by 4.18%, 26.18% 23.9%, and 1.98% for fashion-MNIST, CIFAR10, Breast Cancer, and Brain Tumor datasets. This desirable performance required for complex image (e.g., medical image) diagnosis by the proposed model is attributed to the good feature extraction capabilities (texton and dense layers) that were employed to extract only relevant information from the various images subjected to the model. 

Reviewer 1, Comment # 6: There may be a need to find the statistical significance of the claimed improvement of this work

Response: Thank you for the review. In the revised manuscript, the statistical significance of the improvement made has been stated clearly. [see page: 14-15] 

Author action:

The proposed model’s performance is assessed with that of the traditional CapsNet [28]. The accuracy and loss graphs for the proposed and baseline CapsNet models are displayed in Fig 7. It is clear from the accuracy graphs that the proposed model outperforms the baseline CapsNet model in terms of accuracy and early convergence. This means that the proposed model predicts correctly more than the baseline model. In comparison to the proposed model validation accuracy, which is greater and consistent, the baseline validation accuracy for breast cancer experienced lower accuracy with significant swings. Also, contrary to the proposed model, the baseline CapsNet model for CIFAR-10 exhibits a steady fall in test accuracy following convergence. 

Fig 8 displays the ROC and area under the curve graphs for fashion-MNIST, CIFAR-10, Breast Cancer, and Brain Tumor. The proposed model outperformed the baseline model when the values for the ROC and Precision-Recall curves were taken into account. This shows that, the proposed model is more robust.

The confusion matrices, which are depicted in Fig 9, indicate how many images were correctly and incorrectly identified. We calculated the true positive, false positive, true negative, false negative, accuracy per class values, specificity, sensitivity, and precision to extract relevant information from these figures. In the proposed model, fewer images were incorrectly identified than in the baseline CapsNet model, as shown in Fig 9. It is clear from the true positive and accuracy numbers for each class that the suggested model performed well across all of the classes. [see page: 14]

For the accuracy per class using the breast cancer dataset, for both classes 0 and 1 using the baseline model, 28.89% of the images belonging to these classes were wrongly classified as compared to the proposed model that only misclassified 4.99% of the images belonging to these classes. For the brain tumor dataset, misclassification for classes 0, 1, 2, and 3 using the baseline model was 3.43%, 4.04%, 0.76%, and 0.3% respectively, whiles the proposed model misclassifications were 1.45%, 1.75%, 0.91%, and 0.45% respectively for the same classes. This shows that the proposed model classifies images belonging to the various classes better than the baseline model. Also, for precision using the breast cancer dataset, for both classes 0 and 1 using the baseline model, 27.59% and 28.96% of images predicted to belong to classes 0 and 1 respectively do not belong to those classes, compared to the proposed model, where only 8.71% and 3.2% images for classes 0 and 1 respectively does not belong to those classes. For the brain tumor, using the baseline model, 5.57%, 9.58%, 1.99%, and 0.98% of images for classes 0, 1, 2, and 3 respectively, did not actually belong to these classes compared to 2.05%, 2.99%, 2.94%, and 1.46% of images for classes 0, 1, 2, and 3 respectively, when using the proposed model. It can be seen that the proposed model had fewer images predicted that do not belong to their respective classes than the baseline model. Again, for sensitivity using the breast cancer dataset, for both classes 0 and 1 with the baseline model, 87.3% and 2.21% images belonging to these classes are missed, compared to 7.06% and 4.05% of images missed for classes 0 and 1 respectively using the proposed model. For the brain tumor dataset, using the baseline model, 9.67%, 7.52%, 1.33%, and 0% of images for classes 0, 1, 2, and 3 respectively are missed as compared to 4.33%, 4.57%, 1%, and 0% of images for classes 0, 1, 2, and 3 respectively, missed using the proposed model. This shows that, the proposed model did not miss many images belonging to the various classes, but correctly identified images belonging to particular classes as opposed to the baseline model. Also, for specificity using the breast cancer dataset, for both classes 0 and 1 with the baseline model, 2.21% and 87.3% of images were misclassified as belonging to the various classes respectively, whiles the proposed model misclassified 4.05% and 7.06% images for the various classes respectively as belonging to those classes, whiles they do not belong to them. For the brain tumor, using the baseline model, 1.58%, 2.98%, 0.59%, and 0.44% of images for classes 0,1,2,3 respectively, were misclassified as belonging to the various classes, compared to 0.59%, 0.89%, 0.89%, and 0.66% images for classes 0,1,2,3 respectively, that were misclassified to belong to the various classes by the proposed model whiles they did not belong to them. These figures obtained, show that the proposed model was able to exclude images that do not belong to a particular class better than the baseline model. All this can be found in Table 1 and Table 2., whose values were retrieved from confusion matrices found in Fig 9. Furthermore, the proposed model gives better ROC values than the baseline model, which indicates that the proposed model is capable of distinguishing between classes at various thresholds. Also, the precision-recall values for the proposed model were better, compared to that of the baseline model as shown in Fig 8. These graphs help to access a model’s robustness, when the dataset is imbalanced as can be found in the breast cancer dataset. Overall, the proposed model showed very promising results in terms of accuracy, reliability, and likelihood of false positive and negative predictions, compared to the baseline model, which is a very needed property, in medical image diagnosis. These promising results can be attributed to the good feature extraction capabilities (texton and dense layers) that were employed to extract only relevant information from the various images subjected to the model. [see page: 15]

Reviewer 1, Comment # 5: The result and discussion section are poorly written. The authors need to explain or link the finding with their proposed contributions and how it differs from what is in the available literature.

Response: Thank you for the review. In the revised manuscript, the result and discussion section has been rewritten well to link the finding with the proposed contribution and how it differs from what is available in the literature. [see page: 20]

Author action:

As shown in Table 4, considering the fashion-MNIST dataset, the proposed model that employed texton and dense layers for better feature extraction obtained the highest validation accuracy of 94.90%, followed by Abra and colleagues’ proposed CapsNet model [45] that used Gabor and preprocessing blocks for feature extraction and the least validation accuracy of 90.26% was obtained from Ozan and colleagues proposed model [43] that used quaternions to design a CapsNet model. For CIFAR-10 dataset, a CapsNet model by Phaye and colleagues [41], that used dense blocks for feature extraction achieved the highest validation accuracy of 89.71, followed by the proposed model that achieved a validation accuracy of 89.09%, and the least validation accuracy is from the baseline CapsNet by Sabour and colleagues [28], which uses a convolution layer for feature extraction. For the breast cancer dataset, the proposed model had the highest validation accuracy of 95.01% as compared to 71.11% of the basline model. Lastly, for the Brain tumor dataset, the proposed model had the highest validation accuracy of 97.71% followed by the baseline model at 95.73%, and then the model proposed by Phaye and colleagues obtained the least validation accuracy of 95.03% [41]. The proposed model performs commensurately well compared to the various state-of-the-art capsule network models. The proposed model outperformed the baseline model by 4.18%, 26.18% 23.9%, and 1.98% for fashion-MNIST, CIFAR10, Breast Cancer, and Brain Tumor datasets. This desirable performance required for complex image (e.g., medical image) diagnosis by the proposed model is attributed to the good feature extraction capabilities (texton and dense layers) that were employed to extract only relevant information from the various images subjected to the model. 

Reviewer 1, Comment # 7: The conclusion may need to be improved, relating it to the contributions and findings. Also, state some limitations

Response: Thank you for the review. In the revised manuscript, the conclusion has been improved, relating it to the contributions and findings. Also, some limitations have been stated. [see page: 20-21]

Author action:

An improved Capsule Network model with a texton detection method and dense layers was proposed in this paper to help extract key features and increase the recognition rate on complex images. The adoption of texton detection method and dense layers helped to improve the model’s feature extraction abilities, thus helping to prevent overfitting due to class imbalance, acquiring competitive validation accuracies, and obtaining performances that are appreciable. The proposed model outperformed the CapsNet baseline model and performed comparatively well with the state-of-the-art models in literature on the four datasets in terms of convergence, accuracy, flexibility, robustness, and complexity. When tested against the fashion-MNIST, CIFAR-10, breast cancer, and brain tumor datasets, the suggested model performed admirably. In general, the results of this study clarify the viability of using Capsule Networks on complex tasks in the real world. However, the routing algorithm used by the capsule network has been blamed for decreasing its expressivity as well as hindering it from input distinguishing, which limits the effective classification of the model. In the future, a new routing algorithm coupled with fewer model parameters will be considered to improve the routing of vectors from lower-level to higher-level capsules for better classification and feasible implementation on smartphones.

Reviewer #2: - Spell out each acronym the first time used in the Abstract (CNN) as well as the body of the paper

Response: Thank you for the review. In the revised manuscript, each acronym that was not spelled out the first time has been spelled out [see page: 1] [see page: 2]

Author action:

Convolutional Neural Networks (CNNs) [see page: 1]

Magnetic Resonance Imaging (MRI) [see page: 2]

Reviewer 2, Comment # 1: The author should discuss more recent references in the introduction as below.

*Gaze Prediction Based on Convolutional Neural Network

*Radiologists versus Deep Convolutional Neural Networks: A Comparative Study for Diagnosing COVID-19

*Accurate and compact convolutional neural network based on stochastic computing

*Automated Diagnosis of Acne and Rosacea using Convolution Neural Networks

*Deterministic Modeling to Predict the Natural Gas Density Using Artificial Neural Networks

Response: Thank you for the review. In the revised manuscript, the introduction has been improved based on the recommended paper titled “Automated Diagnosis of Acne and Rosacea using Convolution Neural Networks”. Although the rest of the recommended papers are good, they do not directly relate to the focus and the application area of the study. To improve the introduction of this study authors have also discussed other recent works related to the study. [see page: 2]

Author action: 

CNNs are well-known and frequently used deep learning models because of their propensity to learn relevant and hierarchical features through their feature extraction technique with convolution structures [12]. They have been applied in the medical field, in diagnosing diseases such as Acne and Rosacea [13–15], brain tumors [16–18], breast cancer [19–21], etc. 

Reviewer 2, Comment # 2: The author should improve figures 1 and 2. background of all images should be white.

Response: Thank you for the review. In the revised manuscript figure 1 and 2 has been improved and the background of all images has been made white. [see page: 8, 9] 

Author action:

Figure 1 [see page: 8]

Figure 2 [see page: 9]

Reviewer 2, Comment # 3: Authors should remove title above the Figures 7 and 8. if the same as the caption otherwise it can be written as the caption. and also, background

Response: Thank you for the review. In the revised manuscript, the titles of Figures 7 and 8 have been removed. [see page: 16, 17]

Author action: 

Figure 7: [see page: 16]

Figure 8: [see page: 17]

Reviewer 2, Comment # 4: graphs of figure 7 and 8 are not looking good. author should improve the quality.

Response: Thank you for the review. In the revised manuscript the quality of Figures 7 and 8 have been improved by the authors. [see page: 16] [see page: 17]

Author action:

Figure 7: [see page: 16]

Figure 8: [see page: 17]

Reviewer 2, Comment # 5: Conclusion to be made more systematic, and future scope to be elaborated more on technical features that are planned to be added in the proposed system in the near future.

Response: Response: Thank you for the review. In the revised manuscript, the conclusion has been improved to be more systematic, and the future scope has been elaborated.

Author action:

An improved Capsule Network model with a texton detection method and dense layers was proposed in this paper to help extract key features and increase the recognition rate on complex images. The adoption of texton detection method and dense layers helped to improve the model’s feature extraction abilities, thus helping to prevent overfitting due to class imbalance, acquiring competitive validation accuracies, and obtaining performances that are appreciable. The proposed model outperformed the CapsNet baseline model and performed comparatively well with the state-of-the-art models in literature on the four datasets in terms of convergence, accuracy, flexibility, robustness, and complexity. When tested against the fashion-MNIST, CIFAR-10, breast cancer, and brain tumor datasets, the suggested model performed admirably. In general, the results of this study clarify the viability of using Capsule Networks on complex tasks in the real world. However, the routing algorithm used by the capsule network has been blamed for decreasing its expressivity as well as hindering it from input distinguishing, which limits the effective classification of the model. In the future, a new routing algorithm coupled with fewer model parameters will be considered to improve the routing of vectors from lower-level to higher-level capsules for better classification and feasible implementation on smartphones.

Reviewer 2, Comment # 6: In the Conclusion, future directions and challenges should be explained more.

Response: Thank you for the review. In the revised manuscript, future directions and challenges are explained more in the conclusion [see pages: 20-21]

Author action:

An improved Capsule Network model with a texton detection method and dense layers was proposed in this paper to help extract key features and increase the recognition rate on complex images. The adoption of texton detection method and dense layers helped to improve the model’s feature extraction abilities, thus helping to prevent overfitting due to class imbalance, acquiring competitive validation accuracies, and obtaining performances that are appreciable. The proposed model outperformed the CapsNet baseline model and performed comparatively well with the state-of-the-art models in literature on the four datasets in terms of convergence, accuracy, flexibility, robustness, and complexity. When tested against the fashion-MNIST, CIFAR-10, breast cancer, and brain tumor datasets, the suggested model performed admirably. In general, the results of this study clarify the viability of using Capsule Networks on complex tasks in the real world. However, the routing algorithm used by the capsule network has been blamed for decreasing its expressivity as well as hindering it from input distinguishing, which limits the effective classification of the model. In the future, a new routing algorithm coupled with fewer model parameters will be considered to improve the routing of vectors from lower-level to higher-level capsules for better classification and feasible implementation on smartphones.

Reviewer 2, Comment # 7: please give a proofread check to the paper.

Response: Thank you for the review. In the revised manuscript, thorough proofreading has been done by the autho

---

## [Decision Letter · Decision Letter 1]

5 Sep 2023

PONE-D-23-06111R1TTDCapsNet: Tri Texton-Dense Capsule Network for Complex image recognitionPLOS ONE

Dear Dr. Akoto-Adjepong,

Thank you for submitting your manuscript to PLOS ONE. After careful consideration, we feel that it has merit but does not fully meet PLOS ONE’s publication criteria as it currently stands. Therefore, we invite you to submit a revised version of the manuscript that addresses the points raised during the review process.

Your manuscript has been evaluated by three new reviewers (3, 4 and 5) and their comments are appended below. The reviewers have raised concerns regarding the experiments reported, specifically requesting comparisons to state-of-the-art models, as well as some points regarding clarity of reporting of the performance evaluation and proposed methods. Please ensure you address each of the reviewers' comments when revising your manuscript. Please also note that PLOS ONE has specific guidelines on code sharing for submissions in which author-generated code underpins the findings in the manuscript. In these cases, all author-generated code must be made available without restrictions upon publication of the work. Please review our guidelines at https://journals.plos.org/plosone/s/materials-and-software-sharing#loc-sharing-code and ensure that your code is shared in a way that follows best practice and facilitates reproducibility and reuse.

We look forward to receiving your revised manuscript.

Kind regards,

Hugh Cowley

Staff Editor

PLOS ONE

Reviewers' comments:

Reviewer's Responses to Questions

**Comments to the Author**

1. If the authors have adequately addressed your comments raised in a previous round of review and you feel that this manuscript is now acceptable for publication, you may indicate that here to bypass the “Comments to the Author” section, enter your conflict of interest statement in the “Confidential to Editor” section, and submit your "Accept" recommendation.

Reviewer #1: All comments have been addressed

Reviewer #2: (No Response)

Reviewer #3: (No Response)

Reviewer #4: (No Response)

Reviewer #5: All comments have been addressed

2. Is the manuscript technically sound, and do the data support the conclusions?

Reviewer #1: (No Response)

Reviewer #2: (No Response)

Reviewer #3: (No Response)

Reviewer #4: Partly

Reviewer #5: Yes

3. Has the statistical analysis been performed appropriately and rigorously? 

Reviewer #1: (No Response)

Reviewer #2: (No Response)

Reviewer #3: (No Response)

Reviewer #4: Yes

Reviewer #5: Yes

4. Have the authors made all data underlying the findings in their manuscript fully available?

Reviewer #1: (No Response)

Reviewer #2: (No Response)

Reviewer #3: (No Response)

Reviewer #4: Yes

Reviewer #5: Yes

5. Is the manuscript presented in an intelligible fashion and written in standard English?

Reviewer #1: (No Response)

Reviewer #2: (No Response)

Reviewer #3: (No Response)

Reviewer #4: Yes

Reviewer #5: Yes

6. Review Comments to the Author

Reviewer #1: (No Response)

Reviewer #2: (No Response)

Reviewer #3: From the experimental data, it can be seen that the proposed model lacks effective innovation. The authors can conduct comparative experiments with models from recent years to demonstrate the advantages of the proposed model.

Reviewer #4: The authors proposed a CapsNet-based technique suitable for complex image classification. The abstract and introduction are well-written. Moreover, the contribution of the study is clearly indicated. The literature survey is comprehensive. The methodology is well-written, and the proposed technique is properly explained. The authors can improve the paper as follows:

1. One of the key advantages of CapsNet over CNN is its robustness to image transformations. To further demonstrate the robustness of the proposed technique, the authors can compare the proposed model with state-of-the-art CNN-based architecture such as DenseNet, ResNet, etc. It will be interesting to see which of the algorithms are more robust to image transformation.

2. The authors reported the complexity of the proposed technique. They should also compare the complexity of the proposed technique to that of baseline CapsNet. Which is more optimal? Which is more time-consuming?

3. Why did the authors scale the images in the brain cancer dataset to 32x33x3? What was the original image size? The scaling will affect the image quality and subsequently affect the performance of the model.

4. The authors presented a comparative performance analysis between the proposed technique and baseline CapsNet. Before comparing the two algorithms, the authors should also provide a detailed performance analysis of the proposed technique. They should discuss the performance using all performance metrics presented in the paper. The authors should report and clearly explain the results.

5. The authors should compare the proposed technique and the baseline CapsNet based on their robustness to image transformation, overfitting, generalization performance, etc.

6. If I am interpreting Figure 7a correctly, there seems to be a substantial difference between the training and validation accuracy. If I am correct, the proposed model will not generalize well on unseen images. Kindly provide a discussion of the generalization performance of the proposed model.

Reviewer #5: 1、This manuscript is a detailed and rigorous revision of the previously mentioned comment.

2、In the "Proposed model" section, the reference to "Fig 5. Architecture of the proposed CapsNet model." is not shown in the paper， so please check it and add it in time.

3、In the"Performance evaluation "section, the confusion matrix presented in Fig 9, shows the number of correctly and incorrectly categorized images. In order to better understand the results of the confusion matrix, it is recommended to provide an explanation of each element in the confusion matrix.

4、In the Capsule network section of Proposed methods, there is a lack of detailed explanations for many specialized knowledge, such as the principles of the routing algorithms, how the coupling coefficients are calculated, the protocols between the capsules, and the details of how to update the coupling coefficients.

5、In the experimental section, comparative experiments and ablation experiments are conducted, and It is rigorous and scientific in explaining the results and evaluation indexes in detail.

7. PLOS authors have the option to publish the peer review history of their article (what does this mean?). If published, this will include your full peer review and any attached files.

Reviewer #1: No

Reviewer #2: No

Reviewer #3: No

Reviewer #4: **Yes: **Andronicus Akinyelu

Reviewer #5: No

---

## [Author Response · Author response to Decision Letter 1]

13 Sep 2023

thanks for the review. The revised manuscript,revised manuscript with track changes, and response to reviewers, figures, latex files, and other supporting files have been added.

---

## [Decision Letter · Decision Letter 2]

19 Dec 2023

PONE-D-23-06111R2TTDCapsNet: Tri Texton-Dense Capsule Network for Complex image recognitionPLOS ONE

Dear Dr. Akoto-Adjepong,

Thank you for submitting your manuscript to PLOS ONE. After careful consideration, we feel that it has merit but does not fully meet PLOS ONE’s publication criteria as it currently stands. Therefore, we invite you to submit a revised version of the manuscript that addresses the points raised during the review process.

Your manuscript has been reviewed by three reviewers: two new reviewers, and one of the previous reviewers (Reviewer 5); their comments are appended below. The reviewers (including Reviewer 7, who had no additional comments) are satisfied with the changes you have made in response to the previous reviewer comments. One reviewer has suggested further additions to the list of references; as always, we recommend that you please review and evaluate the requested works to determine whether they are relevant and should be cited. It is not a requirement to cite these works. We appreciate your attention to this request. Additionally, in our previous letters we have stated that PLOS ONE has specific guidelines on code sharing for submissions in which author-generated code underpins the findings in the manuscript. In these cases, all author-generated code must be made available without restrictions upon publication of the work. Please review our guidelines at https://journals.plos.org/plosone/s/materials-and-software-sharing#loc-sharing-code and ensure that your code is shared in a way that follows best practice and facilitates reproducibility and reuse. Compliance with this policy will be a requirement for final publication of your manuscript.

We look forward to receiving your revised manuscript.

Kind regards,

Hugh Cowley

Staff Editor

PLOS ONE

Journal Requirements:

Reviewers' comments:

Reviewer's Responses to Questions

**Comments to the Author**

1. If the authors have adequately addressed your comments raised in a previous round of review and you feel that this manuscript is now acceptable for publication, you may indicate that here to bypass the “Comments to the Author” section, enter your conflict of interest statement in the “Confidential to Editor” section, and submit your "Accept" recommendation.

Reviewer #5: All comments have been addressed

Reviewer #6: All comments have been addressed

Reviewer #7: (No Response)

2. Is the manuscript technically sound, and do the data support the conclusions?

Reviewer #5: Yes

Reviewer #6: Yes

Reviewer #7: (No Response)

3. Has the statistical analysis been performed appropriately and rigorously? 

Reviewer #5: Yes

Reviewer #6: Yes

Reviewer #7: (No Response)

4. Have the authors made all data underlying the findings in their manuscript fully available?

Reviewer #5: Yes

Reviewer #6: Yes

Reviewer #7: (No Response)

5. Is the manuscript presented in an intelligible fashion and written in standard English?

Reviewer #5: Yes

Reviewer #6: Yes

Reviewer #7: (No Response)

6. Review Comments to the Author

Reviewer #5: 1、This manuscript has been comprehensively revised and improved based on previously proposed suggestions.

2、To make the literature study comprehensive, the authors may refer more similar researches in the final publication version, for example, “Multi-lane capsule network for classifying images with complex background” “A novel dense capsule network based on dense capsule layers”.

Reviewer #6: The manuscript proposed a Capsule Network architecture called Tri Texton-Dense CapsNet (TTDCapsNet) for

better complex image classification. The manuscript is carefully written and have addressed all issues raised.

Reviewer #7: (No Response)

7. PLOS authors have the option to publish the peer review history of their article (what does this mean?). If published, this will include your full peer review and any attached files.

Reviewer #5: No

Reviewer #6: No

Reviewer #7: No

---

## [Author Response · Author response to Decision Letter 2]

22 Dec 2023

Reviewer #5: Comment 1: This manuscript has been comprehensively revised and improved based on previously proposed suggestions.

Response: Thank you for the review. 

Reviewer #5: Comment 2: To make the literature study comprehensive, the authors may refer to more similar research in the final publication version, for example, “Multi-lane capsule network for classifying images with complex background” “A novel dense capsule network based on dense capsule layers”.

Response: Thank you for the review. In the revised manuscript, the literature is made comprehensive by referring to more similar research and then used for comparative analysis. [see pages: 5 and 18]

Author action: 

 Chang and team innovatively crafted a capsule-oriented network architecture, introducing modifications to the Squash function and refining the dropout implementation. Thorough experimentation was undertaken on three widely-used datasets—MNIST, affNIST, and CIFAR10—yielding accuracies of 99.73%, 81.71%, and 76.79%, respectively [98]. Also, Sun and colleagues introduced an innovative dense capsule network called DenseCaps, which utilizes dense capsule layers. The architecture comprises three Dense Capsule Blocks to facilitate the reuse of features and multi-scale feature capsule processing calculation of loss. The model demonstrated notable performance with an accuracy of 94.93% on fashion-MNIST and 89.41% on CIFAR-10 [99]. Through the incorporation of the Convolutional Capsule Layer (Conv-Caps Layer), Xiong and team enhanced CapsNet, leading to a significant boost in its overall performance. This enhancement resulted in validation accuracies of 81.29% for CIFAR-10 and 99.84% for MNIST datasets [100]. Do and colleagues presented a multi-lane capsule network (MLCN) designed for parallel processing, delivering notable accuracy with reduced costs. The MLCN comprises distinct parallel lanes, each contributing to a specific result dimension. Evaluation of the model on fashion-MNIST and CIFAR-10 datasets demonstrated validation accuracies of 92.63% and 75.18%, respectively [101]. CHENG and colleagues introduced the Complex-valued Diverse CapsNet (Cv-CapsNet++), which involves a three-stage process. Initially, a restricted dense complex-valued subnetwork is utilized to acquire multi-scale complex-valued features. Subsequently, these features are encoded into complex-valued primary capsules in the second stage. Lastly, in the third stage, they extended the dynamic routing algorithm to the complex-valued domain, employing it to integrate real- and imaginary-valued information from the complex-valued primary capsules. The model achieved validation accuracies of 94.40% on Fashion-MNIST and 86.70% on CIFAR-10 datasets [102]. Shiri and her team substituted the second convolutional layer with a fully connected (FC) layer and directed the result of the initial convolutional layer into another FC layer. The FC layer's output was subsequently transformed to generate Primary Capsules, serving as inputs for the dynamic routing algorithm. Notably, the model attained validation accuracies of 88.84% and 67.18% on the fashion-MNIST and CIFAR-10 datasets, respectively [103]. Huang and Zhou introduced a capsule network with a dual attention mechanism, referred to as DA-CapsNet. In this model, the initial attention layer follows the convolution layer, while the subsequent attention layer is incorporated after the primary capsule layer. The model demonstrated notable validation accuracies of 93.98% and 85.47% on the fashion-MNIST and CIFAR-10 datasets respectively [104]. Luo and his team introduced R-CapsNet, drawing inspiration from the impact of a compact convolution kernel. This design serves as an expansion of the initial CapsNet, featuring four convolutional layers and one fully connected layer. During testing on the fashion-MNIST and CIFAR-10 datasets, the model demonstrated validation accuracies of 93.89% and 81.2% respectively [105]. [see page: 5]

Methods Validation accuracy (%)

 fashion-MNIST CIFAR 10 Breast Cancer Brain Tumor

ShallowNet [33] 92.70 75.75 

CapsNet [28] 90.72 62.91 71.11 95.73

Enhanced-CapsNet [34] - 82.31 - -

64 Capsule Layers [31] - 64.67 - -

Feature Amplification CapsNet [35] 93.76 84.56 - -

Multi-lane [32] 92.63 76.79 - -

MS-CapsNet [36] 92.70 75.70 - -

ResCapsNet [37] - 78.54 - -

CFC-CapsNet [38] 92.86 73.15 - -

Fast Inference [39] 91.52 70.33 - -

Inverted dot product [40] - 82.55 - -

DCNET++ [41] 94.65 89.71 - 95.03

Max–min [42] 92.07 75.92 - -

Quaternion CapsNet [43] 90.26 82.21 - -

Gabor capsule network with prep blocks [45] 94.78 85.24 - -

SqueezeCapsNet [93] 93.49 82.45 - -

Afriyie et al [94] 92.8 75.42 - -

MCNet [95] 93.17 79.27 - -

Afriyie et al [96] 94.93 84.57 - -

MLSCN [98] - 76.79 - -

DenseCaps [99] 94.93 89.41 - -

DeeperCaps [100] - 81.29 - -

MLCN [101] 92.63 75.18 - -

Cv-CapsNet++ [102] 94.40 86.70 - -

Quick-CapsNet (QCN) [103] 88.84 67.18 - -

DA-CapsNet [104] 93.98 85.47 - -

R-CapsNet [105] 93.89 81.57 - -

Proposed Model 94.90 89.09 95.01 97.71

[see page: 18]

Reviewer #6: The manuscript proposed a Capsule Network architecture called Tri Texton-Dense CapsNet (TTDCapsNet) for better complex image classification. The manuscript is carefully written and has addressed all issues raised.

Response: Thank you for the review. 

Reviewer #7: (No Response)

Response: Thank you for the review.

---

## [Editor Report · Decision Letter 3]

25 Jan 2024

PONE-D-23-06111R3TTDCapsNet: Tri Texton-Dense Capsule Network for Complex image recognitionPLOS ONE

Dear Dr. Akoto-Adjepong,

Thank you for submitting your manuscript to PLOS ONE. After careful consideration, we feel that it has merit but does not fully meet PLOS ONE’s publication criteria as it currently stands. Therefore, we invite you to submit a revised version of the manuscript that addresses the points raised during the review process.

We look forward to receiving your revised manuscript.

Kind regards,

Hugh Cowley

Staff Editor

PLOS ONE

Additional Editor Comments:

We note that you do not appear to have responded to the previously stated PLOS ONE requirements regarding code sharing.

PLOS ONE has specific guidelines on code sharing for submissions in which author-generated code underpins the findings in the manuscript. In these cases, all author-generated code must be made available without restrictions upon publication of the work.

Please review our guidelines at https://journals.plos.org/plosone/s/materials-and-software-sharing#loc-sharing-code and ensure that your code is shared in a way that follows best practice and facilitates reproducibility and reuse.

Please note that we will not publish your manuscript unless these requirements have been met.

---

## [Author Response · Author response to Decision Letter 3]

3 Feb 2024

code that underpins the findings in the manuscript has been publicly made available without restrictions. This can be found under Data Availability; https://github.com/vivianakotoadjepong/TTDCapsNet.git

---

## [Editor Report · Decision Letter 4]

22 Feb 2024

TTDCapsNet: Tri Texton-Dense Capsule Network for Complex and Medical Image Recognition

PONE-D-23-06111R4

Dear Dr. Akoto-Adjepong,

We’re pleased to inform you that your manuscript has been judged scientifically suitable for publication and will be formally accepted for publication once it meets all outstanding technical requirements.

Kind regards,

Hugh Cowley

Staff Editor

PLOS ONE
---

## [Editor Report · Acceptance letter]

6 Mar 2024

PONE-D-23-06111R4 

PLOS ONE

Dear Dr. Akoto-Adjepong, 

I'm pleased to inform you that your manuscript has been deemed suitable for publication in PLOS ONE. Congratulations! Your manuscript is now being handed over to our production team.

Kind regards, 

on behalf of

Mr Hugh Cowley 

Staff Editor

PLOS ONE